# Sequenced Combinations of Cisplatin and Selected Phytochemicals towards Overcoming Drug Resistance in Ovarian Tumour Models

**DOI:** 10.3390/ijms21207500

**Published:** 2020-10-12

**Authors:** Safiah Ibrahim Althurwi, Jun Q. Yu, Philip Beale, Fazlul Huq

**Affiliations:** 1School of Medical Sciences, University of Sydney, Sydney NSW 2006, Australia; sssafy2008@hotmail.com (S.I.A.); jun.yu@sydney.edu.au (J.Q.Y.); 2Department of Medical Oncology, Concord Repatriation General Hospital, Concord NSW 2137, Australia; Philip.beale@health.nsw.gov.au; 3Eman Research Ltd., Canberra ACT 2609, Australia

**Keywords:** ovarian cancer, drug resistance, apoptosis, proteomics, combination, cytotoxicity, artemisinin, oleanolic acid, platinum drugs, cisplatin

## Abstract

In the present study, cisplatin, artemisinin, and oleanolic acid were evaluated alone, and in combination, on human ovarian A2780, A2780^ZD0473R^, and A2780^cisR^ cancer cell lines, with the aim of overcoming cisplatin resistance and side effects. Cytotoxicity was assessed by MTT reduction assay. Combination index (CI) values were used as a measure of combined drug effect. MALDI TOF/TOF MS/MS and 2-DE gel electrophoresis were used to identify protein biomarkers in ovarian cancer and to evaluate combination effects. Synergism from combinations was dependent on concentration and sequence of administration. Generally, bolus was most synergistic. Moreover, 49 proteins differently expressed by 2 ≥ fold were: CYPA, EIF5A1, Op18, p18, LDHB, P4HB, HSP7C, GRP94, ERp57, mortalin, IMMT, CLIC1, NM23, PSA3,1433Z, and HSP90B were down-regulated, whereas hnRNPA1, hnRNPA2/B1, EF2, GOT1, EF1A1, VIME, BIP, ATP5H, APG2, VINC, KPYM, RAN, PSA7, TPI, PGK1, ACTG and VDAC1 were up-regulated, while TCPA, TCPH, TCPB, PRDX6, EF1G, ATPA, ENOA, PRDX1, MCM7, GBLP, PSAT, Hop, EFTU, PGAM1, SERA and CAH2 were not-expressed in A2780^cisR^ cells. The proteins were found to play critical roles in cell cycle regulation, metabolism, and biosynthetic processes and drug resistance and detoxification. Results indicate that appropriately sequenced combinations of cisplatin with artemisinin (ART) and oleanolic acid (OA) may provide a means to reduce side effects and circumvent platinum resistance.

## 1. Introduction

Ovarian cancer is one of the major causes of death in gynaecologic cancers [1] marked with absence of reliable screening methods and distinctive symptoms [2,3]. Although widely used, efficacy of platinum drugs, cisplatin [1], oxaliplatin, and carboplatin, is limited due to acquired resistance and adverse side effects. Combination treatments [4] with tumour active plant based compounds that have atypical mechanisms of action, and minor or no side effects, can provide a means to overcome drug resistance [5]. Resistance to platinum-based drugs [6,7] is related to four major mechanisms, which are: (1) increased detoxification and degradation (such as by glutathione); (2) reduced drug accumulation (due to increased efflux and/or reduced influx); (3) increased repair of the Pt-DNA adducts; or (4) enhanced tolerance of the damaged DNA [8].

Oleanolic acid (OA) is a natural pentacyclic triterpenoid (Figure 1) [9], predominantly found in olive oil [10], which exhibits antioxidant, anti-inflammatory, anticancer, and hepatoprotective activities [9]. OA is reported to inhibit metastasis, angiogenesis, and invasion in different cancers [10], while it induces programmed cell death in different cancer cell lines, such as osteosarcoma, liver, gastric, breast, prostate, pancreatic, colorectal, and bladder cancer cells [10] via enhancement of p38 MAPK, ASK1, ROS, and inhibition of signalling pathways, such as S6K, PI3K, mTOR, Akt, and NF-κB in time- and concentration-dependent fashion [10]. However, it shows pro-apoptotic and anti-proliferative effects through down-regulation of survivin, Bcl-xL, Bcl-2, and other anti-apoptosis proteins accompanied with modulation of MRP1 activity [10]. Artemisinin (ART), is obtained from (Sweet wormwood, *Artemisia annua* L), (Figure 1) [11]. It inhibits metastasis, invasion, and angiogenesis, while it induces apoptosis and cell cycle arrest [11]. Artemisinin and its derivatives also exhibit anti-malarial, and anti-inflammatory activities [11] through modulation of molecular pathways, such as tyrosine kinases, STAT-1/3/5, NF-κB, MAPKs, PI3K/Akt Sp1, Toll-like receptors, Nrf2/ARE phospholipase, p53, and MDM2 oncogene [11]. Another study reported that artemisinins increase E-cadherin, Bax and reduce MMPs, BCL-2, HIF-1α, VEGF, CDKs, and cyclins [12]. It has been suggested that ART induces DNA damage and apoptosis by producing free radicals that result from the reaction of its endoperoxide moiety with intracellular free iron [12]. However, the effect is influenced by cancer cell phenotype and the origin of tissue [13]. A growing body of evidence suggests that the combination treatment with plant-based anticancer agents and common cancer drugs can improve effectiveness of chemotherapeutic agents through regulation of pathways, including the ones regulated by COX-2, nuclear factor-KB, and Akt (which are also associated with drug resistance) [5]. Thus, it is thought that artemisinin and oleanolic acid would act synergistically with cisplatin by regulating various pathways and repressing different mechanisms involved in cancer initiation, metastasis, invasion, and angiogenesis.

The current study aimed to examine efficacy from binary combinations of artemisinin and oleanolic acid with cisplatin in A2780, A2780^cisR^ (A2780 made resistant to cisplatin), and A2780^ZD0473R^ (A2780 made resistant to platinum compound ZD0473) human ovarian cancer cell lines. Selected combinations were chosen to determine their effect on cellular platinum accumulation and Pt−DNA binding levels, changes in protein expression (using 2-DE gel electrophoresis). Finally, MALDI -TOF-TOF MS/MS was used to identify the proteins. Various studies relate to drug efficacy alone and in combination, and the mechanisms of action.

## 2. Results

### 2.1. Growth-Inhibitory Effect of Single Drugs

The concentration of cisplatin (CS), OA, and ART that caused 50% growth inhibition in A2780, A2780^cisR^, and A2780^ZD0473R^ cell lines, was identified as IC_50_ values. The data revealed (in Table 1) that the highest resistant factor (RF) value was applied to CS, whereas the lowest was produced by OA. Among the two phytochemicals in the A2780 cell line, ART was the more active, whereas, in the A2780^cisR^ and A2780^ZD0473R^ cell lines, OA was the more active compound.

### 2.2. Combination Studies

Combination indices (CIs) were calculated using the formula developed by Chou and Talalay to determine whether the combined drug action was antagonistic, synergistic, or additive. A CI value < 1 means a synergistic effect, equal to one means an additive effect, and > 1 means an antagonistic effect of the combined drugs [14] ED_50_, ED_75_, and ED_90_ standing for concentrations applying respectively to 50%, 75%, and 90% growth inhibition, were determined for all combinations. In essence, ED_50_, ED_75_, and ED_90_ are equivalent to IC_50_, IC_75_, and IC_90_. D_m_ is the median effect dose and m is the exponent defining shape of the dose response curve. All drugs in combination were added using the following three modes of administration: (0/0 h, 0/4 h, and 4/0 h) to the selected cell lines where 0/0 h, 0/4 h, and 4/0 h respectively mean both drugs are added at the same time; cisplatin was added first, followed by phytochemical four hours later, and phytochemical was added first followed by cisplatin four hours later. In regards to the A2780 cell line, the data show that CS and ART (Table 2 and Figure 2), using 0/0 h produced the highest synergistic effect while the lowest synergistic outcome was produced from (0/4 h). In contrast, CS and OA (Table 3 and Figure 2), using 0/4 h showed the highest synergistic outcome and the least was produced from 4/0 h. On the other hand, in relation to the A2780^cisR^ cell line, CS and ART (Table 3 and Figure 2) using 0/0 h exhibited the highest synergistic outcome, and the lowest resulted from 4/0 h combination. Likewise, CS and OA (Table 3 and Figure 2) using 0/0 h produced the highest synergistic outcome, and the lowest resulted from the 4/0 h combination. Moreover, regarding the A2780^ZD473R^ cell line, CS and ART in combination (Table 4 and Figure 2), using 0/0 h, exhibited the highest synergistic outcome. Likewise, CS and OA in combination (Table 4 and Figure 2), using 0/0 h, resulted in the highest synergistic outcome. The results are discussed fully in the next section. 

### 2.3. Cellular Accumulation of Platinum and Platinum–DNA Binding Level

#### 2.3.1. Cellular Accumulation of Platinum

The data (Table 5 and Figure 3) show that platinum accumulations from the combinations of CS with ART or OA were always greater than those from CS alone in both the cell lines (A2780 and A2780^cisR^), indicating that the presence of ART and OA have served either to increase the uptake or decrease the efflux or both.

#### 2.3.2. Platinum-DNA Binding Level

The data reveal that the platinum–DNA binding levels resulting from the combination of CS with ART or OA (Table 6 and Figure 4) in A2780 were not always higher than those from CS alone, indicating that a positive correlation between increased levels of CS-binding and synergistic outcomes is not always found. For example, the level of CS-DNA binding from the combination of CS with OA using 0/4 h was less than that of CS alone. Moreover, in A2780, CS-DNA binding level was sequence-dependent, but not synergistic in outcome. As the level of CS-DNA binding resulted from combinations of CS with ART using 0/0 h was lowest, whereas combination index showed that 0/0 h sequence was the most synergistic sequence. On the contrary, in the A2780^ciR^ cell line, platinum-DNA binding level from combination treatments was always higher than those from CS alone, suggesting the presence of such a correlation. Moreover, the observed increase in CS-binding levels in the A2780^ciR^ cell line from combination with ART and OA are consistent with the increase in CS-accumulation levels due to the treatments.

**Proteomics** has analysed causes of drug resistance, in terms of the changes in protein expression, and has argued that ART and OA would be able to regulate the expression level of those proteins. Thus, proteins in the A2780 and A2780^cisR^ cell lines were inspected by 2-D gel electrophoresis and mass spectrometry to examine their association with platinum drug resistance, wherein the protein expression levels in the A2780^cisR^ cell line were compared to what was found in its counterpart cell line to determine the proteins that were differentially expressed. The objective of the proteomic study, besides the determination of new ovarian cancer biomarkers, was to evaluate the effect of: CS and ART (using 0/0 h), CS and OA (using 0/4 h), CS and ART (using 0/0 h), and CS and OA (using 0/0 h) on the expression of those dysregulated proteins (differently expressed proteins). Table 7 and Table 8 show protein expression in: (1) untreated A2780^cisR^ as compared to the values found in the untreated A2780 cell line, and (2) treated A2780^cisR^ cell line (with the chosen drug treatments, as mentioned above) as compared to the values that were detected in both untreated cell lines. In both situations, at the beginning, the A2780 cell line was used as a reference. Later, the A2780^cisR^ cell line was used as the reference. Proteins were considered to have undergone significant changes in expression if the fold factor was ≥ 2.0. The proteins that were detected by MALDI TOF/TOF MS and were matched against the Mascot database (http://www.matrixscience.com) and the UniProtKB/Swiss-Prot database (http://www.uniprot.org/) [15], when A2780 and A2780^cisR^ cell lines alternatively were used as a reference, are illustrated in tables (Appendix A and Table 9. Accession number, protein ID, name and mass, MSMS, degree of coverage and score were obtained from APAF (http://www.proteome.org.au/). Theoretical isoelectric point (*pI*), subcellular location, mass spectrum, and matched peptides were obtained from the Mascot database (http://www.matrixscience.com) and Swiss-Prot database (http://www.uniprot.org/) [15] through APAF (http://www.proteome.org.au/). In this study, 49 proteins out of 101 scored above 56, according to the Mascot parameter that protein scores above 56 are considered significant (see Mascot database (http://www.matrixscience.com), Swiss-Prot database (http://www.uniprot.org/) [15], and APAF (http://www.proteome.org.au/). Accordingly, those proteins were investigated for their involvement in the development of platinum resistance, besides evaluating the effect of the selected drug treatments on their expression.

## 3. Discussion

### 3.1. Cytotoxicity of Single Drug

The results show that OA showed greater activity against A2780^cisR^ and A2780^ZD0473R^ cell lines than against parent A2780 cell line, ART showed greater sensitivity towards A2780 and A2780^cisR^ than A2780^ZD0473R^ cells while CS showed lower activities towards A2780^cisR^ and A2780^ZD0473R^ cell lines than the parent cell line. It has been reported that most of the natural products can modify several factors involved in the oncogenic transcription such as; Nrf-2, Hh /GLI, STAT3, FoxM1, PPAR y, Wnt/β-catenin, HGFR, HIF1 α, AP-1 and NF-κB [16]. In contrast, aberrant expression of STAT3, HIF1 α, NF-κB, AP-1 and FoxM1 are often observed in different types of cancers [16]. That OA was found to be more active against A2780^cisR^ and A2780^ZD0473R^ cell lines may be attributed to its apoptotic effect, including the enhancement of p38 MAPK, ASK1 and ROS, and its inhibition effect of signalling pathways such as; S6K, PI3K, mTOR, Akt and NF-κB [10]. These functions make the OA more active towards platinum resistant cells. 

### 3.2. Combination Indices

The results show that the synergistic (Figure 2 and Figure 3 and Table 2, Table 3 and Table 4) effect from the combinations of CS with OA and ART were dependent on the dose and the sequence of administration in the three cell lines. The synergism from combination of OA and ART with CS, however, is not surprising given the fact that OA and ART are known as apoptotic inducers (through increasing Bax, decreasing Bcl-2, and up-regulating of caspases), as well as they exhibit tumour inhibition effect (through the down-regulation of NF-κB and other signalling pathways) and anti-angiogenic effect (through the decreasing of VEGF) [10,17]. Hence, growth inhibition resulting from combination of OA and ART with CS may be attributed to the down regulation of Bcl-2 and NF-κB pathways. Additionally, synergistic outcomes can be ascribed to the effect of ART on cell-cycle that involves suppression of CDK4 and CDK2 expressions [13] as well as inhibition of cyclin A, cyclin E, cyclin D3, cyclin D1, CDK6, E2F1, and JAB1 transcriptions with enhancement of IFIT3, p21, and p27 [17]. Furthermore, it is likely that the significant increase in growth inhibition may be partly attributed to OA’s anticancer effect through the regulation of various signalling cascades, including the inhibition of S6K, PI3K, mTOR, Akt, and NF-κB signalling pathways [10]. The reason why 0/0 h sequence of administration of CS in combinations with ART and OA have generally produced the most synergistic outcomes, may be attributed to the time-dependent nature of the actions of ART and OA. In addition, Ziberna et al. reported that, in different multi-drug resistant cancer cell lines, OA exerts its apoptotic effect via the enhancement of p38 MAPK, ASK1, and ROS pathways in time- and concentration-dependent fashion [10]. Similar to our results, enhanced growth inhibition in HCC was observed from co-administration of OA and sorafenib (chemotherapeutic drug) [10]. It is possible that presence of ART or OA might influence platinum influx and efflux, apoptosis induction, Pt–DNA adducts tolerance, and DNA repair. In support of the idea, it may be noted that Ziberna et al. suggested that intracellular concentration levels of chemotherapeutic drugs were enhanced because of OA’s ability to inhibit the efflux transporters [10]. The combination of OA with 5-fluorouracil also produced synergistic effect and induced apoptosis in pancreatic cancer. Thus, it was suggested that the combining of OA with other chemotherapeutic drugs would produce synergistic anticancer effect [10]. A previous study, using ICP-MS has found differential cellular uptakes in the cisplatin-sensitive ovarian carcinoma A2780 and cisplatin-resistant ovarian carcinoma A2780^cisR^, where the Pt levels in the resistant cell line were found to be significantly lower [18], suggesting, that resistance could be partially attributed to a decrease in the intracellular drug concentration [18]. Another study using ICP-MS to measure platinum levels has also shown that that A2780 cells accumulate more than twofold of platinum than wild-type p53 A2780/CP70 (cisplatin-resistant) cells [19]. Moreover, measuring platinum at the level of a single cell using Single Cell ICP-MS method in ovarian cancer cell lines A2780 and A2780/CP70 shows that the cisplatin uptake is lower in the cisplatin resistant A2780/CP70 cell line [20].

### 3.3. Cellular Accumulation of Platinum

The results showed that Pt accumulations (Table 5 and Figure 2) from combination of CS with the chosen drugs were always greater than that from CS alone, irrespective of nature of the combined drug action. For example, CS in combination with ART and OA using 0/0 h have produced greater Pt accumulations than those from the other sequences in the A2780 cell line. The Pt level from synergistic combination of CS with ART using 0/0 h was 2.9 times higher than the levels from the treatment with (4/0 h) combination of the compounds in the same cell line. Additionally, Pt level from the synergistic combination of CS and ART using 0/4 h, in A2780 cells, was about four times greater from the synergistic 4/0 h combination in the same cell line. Moreover, platinum level resulting from the synergistic 0/0 h combination of CS with ART showed three times higher value than that from 0/4 h combination in A2780^cisR^ cells and 8.5 times higher than that from CS alone. Furthermore, in A2780^cisR^ cells, Pt level resulted from the synergistic 4/0 h combination of CS and ART was nearly two times greater than that from synergistic 0/4 h combination and 4.5 times greater than that from CS alone. Together, these findings provide further support for the hypothesis that the ART growth inhibition action is a sequence and cell type dependent. The results are in agreement with the findings of Gong et al., wherein they reported that ART action was a cell type dependent [13]. On the other hand, synergistic combination of CS with OA using 0/0 h combination in A2780 cells, has increased Pt level to around 15 times greater than that from CS alone, 8.2 and 10 times greater than that from the synergistic sequences of 0/4 h and 4/0 h, respectively. Furthermore, it was found that in the A2780^cisR^ cell line, Pt level from synergistic combination of CS with OA using 0/0 h sequence was nearly two times greater than that from treatment with 0/4 h combination and 3.3 times greater than that from CS alone. Moreover, treatment with synergistic combination 4/0 h, CS accumulation was 1.3 times greater than that resulted from 0/0 h combination and four times greater than that from CS alone. It appears that intracellular accumulation of CS in the presence of OA is also sequence-dependent. Thus, the observed increase in CS accumulation could be attributed to either the ability of OA to increase the influx of CS or to decrease CS efflux or both. The findings are supported by those of Ziberna et al., wherein they reported that intracellular concentration levels of chemotherapeutic drugs were enhanced because of OA’s ability to inhibit the efflux transporters [10]. The inconsistency may be due to the fact that actions of OA are time- and concentration-dependent [10]. 

### 3.4. Platinum–DNA Binding Level

Generally, platinum-DNA levels were greater in the A2780^cisR^ cells than in A278 cells. A possible explanation is that the presence of ART and OA might have modulated some of the platinum resistance mechanisms, such as decreases in platinum-DNA adducts formation [21], increased platinum detoxification, and increased DNA repair [22]. Another possible explanation is that the compounds led to an increase in the platinum uptake so that more platinum was available to bind DNA. Additionally, the compounds could have prevented efflux of the platinum so that the platinum had more chance to cause DNA damages before being deactivated. A combination of CS with ART and OA increased Pt-DNA binding levels, which are consistent with the observed increase in Pt accumulation resulting from the combination of CS with the phytochemicals. Overall, the results presented in (Table 6) showed that with the combinations of CS with ART and OA in the A2780 cell line, platinum-DNA binding levels were not always higher than those of CS alone. For example, in A2780 cell line, platinum-DNA binding levels from synergistic combination of CS with OA using 0/4 h sequence and synergistic combination of CS with ART using 4/0 h sequence were significantly lower than that from CS alone. These results have highlighted the complexity of the situation due to involvement of multiple pathways associated with apoptosis and drug resistance, including the interaction between ART and OA with CS. It can thus be suggested that the variations in the overall outcomes could be attributed to: (1) the dual behaviour of most of the phytochemicals as antioxidant agent (protective effect) at low concentrations and as a pro-oxidant (apoptotic) enhancement at high concentrations; (2) concentration and time dependent effect of the phytochemicals; (3) growth inhibitory effect of platinum drugs as stated previously not being merely dependent on abilities to bind to DNA (although it is an essential step, but not sufficient for the apoptosis induction); (4) ROS formed by platinum drugs, for example cisplatin, is dependent on its concentration and the duration of exposure [18]; thereby, enhancing apoptosis induction [21]. For instance, in this study 24 h incubation period was used as against 72 h for studies on cytotoxicity that may contribute to the overall outcomes. Finally, it has been reported that platinum resistance might have resulted from enhanced tolerance [23], reduction in the adducts formation [18], and enhanced DNA damage repair [24].

### 3.5. In Proteomics Study Involving 2-D Gel Electrophoresis

A total of 133 spots were identified, out of which 101 proteins were significantly altered in expression. Among them, 54 spots were successfully identified by MS. In some cases, mass spectral analysis showed that the same protein was named for more than one spot such as spots 16, 18, and 76 were assigned to hnRNPA2/B1, spots 17 and 31 were assigned to ENOA, spots 52 and 63 were assigned to EF2, and spots 36 and 96 were assigned to VIME. Assignment of different spots to the same protein that exists in multiple isoforms with different *pI*s, might be attributed to the post-translational modifications, which was suggested to occur in 2-D gel electrophoresis-based proteomics studies [24,25], thus, reducing the total to (49) proteins. Based on the protein functions, proteins differentially expressed in the A2780^cisR^ cell line were divided into nine major groups as discussed in the next section, using A2780 and A2780^cisR^ cell lines alternatively as a reference, and are illustrated in Table 10 and Table 11.

*Mortalin*, also known as GRP75 [15], a member of chaperones and stress related proteins, which plays essential function in mitochondrial proteins folding [26], cancer cell proliferation, and enhancement of angiogenesis [27]. In this study, mortalin was found to be down-regulated in the cisplatin-resistant A2780^cisR^ cell line as compared to the sensitive A2780 cell line, consistent with the earlier studies [28,29,30]. The findings indicate that mortalin may play a role drug resistance, representing mortalin as potential diagnostic and therapeutic targets. Treatments with the synergistic combinations of (CS and OA using 0/0 h) and (CS and ART using 0/0 h), partially restored mortalin expression in A2780^cisR^ cell line, suggesting that the applied treatments have improved chaperone action towards greater cell kill.

*APG2* is another molecular chaperone and stress related protein, also known as HSP74 [15], is ubiquitously expressed in different organs [31]. In this study APG2 was over-expressed in the cisplatin-resistant A2780^cisR^ cell line as compared to the sensitive A2780 cell line. Up-regulation of APG2 was previously reported in hepatocellular carcinomas [31]. The elevated expression was related to drug resistance, poor prognosis or advanced stages in different type of cancers [31]. The findings indicate that APG2 plays a role in the development of drug resistance, suggesting that APG2 can be a potential prognostic and therapeutic target. Expression of APG2 was lowered below threshold of detection after treatment with synergistic combinations of (CS and ART using 4/0 h), (CS and OA using 0/4), (CS and ART using 0/0 h) and (CS and OA using 0/0), that might have inhibited its behaviour as heat shock protein exposing cells to the toxic effect of CS, thereby increased cell apoptosis, indicating a causal relationship between APG2 expression and platinum resistance in ovarian cancer. 

*BIP*, also known as GRP78 [15], was over-expressed in a large number of cancers, including malignant gliomas, lung cancer, and breast cancer [32]. It enhances tumour metastasis, survival, proliferation, and resistance to a wide range of therapies [33], including cisplatin [34]. In line with reported findings, in the present study, BIP was up-regulated in the A2780^cisR^ cell line as compared to A2780 cell line. Treatment with selected combinations, over-restored BIP expression, indicating that the treatments have prevented BIP’s roles in cell protection and the tumorigenicity of cancer cells, and thereby increasing the death of cells exposed to CS, suggesting that the selected combinations can serve as alternative therapeutic methods for the resistance in ovarian cancer associated with abnormal BIP expression. Thus, it seems that BIP may possibly be playing a major role in these synergistic outcomes.

*HSP90B*, also known as HS90B [15], is one of the HSP90 isoforms found in mammalian cells [35,36]. HSP90 is involved in the differentiation and cell proliferation processes [37]. It exhibits an anti-apoptotic effect through modulation of NF-κB, TNF, and AKT pathways and enhances tumour growth and angiogenesis through the VEGF pathway [38]. In the present study, HSP90B was down-regulated in the *cis*-resistant cell line as compared to A2780 cell line, consistent with the findings of earlier studies [21,26,27]. The findings indicate that HSP90B plays an important role in apoptosis pathway and in the development of drug resistance. HSP90B expression was partially/over-restored after the treatment with the selected combinations, possibly be due to the effect of the treatments on HSP90B capability to interact with survival signal, including NF-κB and AKT, leading to their inactivation thereby sensitizing cells to apoptosis mediated by the stresses produced by CS. Hence restoration of HSP90B can be a contributing factor responsible for apoptosis enhancement so that the level of its expression can be a means for drug resistance in ovarian cancer. 

*GRP94* also known as ENPL [15], is a member of hsp90 family. The high level of GRP94 was found in many cancers such as breast, gastric, oesophageal, colon, lung, and breast cancer [39]. However, in the present study, GRP94 was down-regulated in the cisplatin-resistant A2780^cisR^ cell line as compared to A2780 cell line. The result is consistent with the findings of earlier studies [24,28,30]. Collectively, the findings suggest that GRP94 expression is cell and cancer dependent and suggest GRP94 as being a potential diagnostic and therapeutic target. Treatment with selected combinations over-restored GRP94 expression. Together with the fact that the GRP94, as one member of HSPs family, protects cells from stressful stimuli, suggests that this protein could play a role in tumorigenicity and drug resistance in ovarian. Thus, the result supports the possibility of a causal relationship between GRP94 expression and the resistance caused by CS.

*CYPA* is one of CyPs isoforms (also known as PPIA) [15]. CYPA catalyses *cis-trans* isomerization of the peptide bonds [15,40]. CYPA is up-regulated in glioblastoma multiforme, melanoma, colorectal, breast and pancreatic cancer [41]. In contrast in the present study, CYPA was down-regulated in the *cis*-resistant cell line as compared to the sensitive A2780 cell line, consistent with the findings of the earlier studies [29,30]. The results suggest that CYPA is involved in drug resistance in ovarian cancer, Treatment with synergistic combinations of (CS and ART using 4/0 h), (CS and OA using 0/4 h) (CS and ART using 0/0 h) partially restored CYPA expression. While after treatment with (CS and OA using 0/0 h) CYPA was not detectable, suggesting that its expression is very sensitive to the treatment. Where the enhanced apoptosis be brought about may be the result of OA’s signalling inhibitory mechanism and its ROS production that have modulated CYPA activities of antioxidant stabilization, transcriptional control, trafficking, cell cycle regulation and signal transduction [42]. Thus, the results suggest that there is a strong correlation between CYPA expression and apoptosis; thus the chosen treatments can serve as potential approaches to overcome drug resistance associated with atypical CYPA expression.

*P4HB* is one member of the PDI family, and is also known as PDIA1 [15]. It has multifunctional activities besides its involvement in the disulphides breakage, formation, and rearrangement at different cellular locations [43]. P4HB is up-regulated in glioblastoma cancer [44]. Silencing of P4HB inhibits tumour survival of MCF-7 cells; however, it did not show a significant inhibitory effect on HeLa cells [44]. Thus, it has been suggested that P4HB expression is a cell-type dependent [44]. In the present study, P4HB was down-regulated in A2780^cisR^ cell line as compared to sensitive A2780 cell line. The result is in accordance with the previous findings of the earlier studies [24,28,30]. The data clearly points to roles of P4HB in cancer progression and resistance, while its expression is cell and cancer-dependent. Treatment with synergistic combinations of (CS and OA using 0/0 h) over-restored P4HB expression. Whereas, treatment with synergistic combinations of (CS and ART using 0/0 h) partially restored P4HB expression, suggesting the involvement of P4HB in apoptotic pathway and indicating that the rise in growth inhibition was caused by the increased P4HB expression. Therefore, the selected combinations are effective in overcoming drug resistance in ovarian cancer model. 

*ERp57* is another member of the PDI family, also known as Erp60, PDIA3, and GRP58 [15], is a chaperone, oxidoreductase and disulphide isomerase protein [45]. ERp57 plays a crucial role along with CRT and CNX chaperones in folding process of disulphide bond-containing proteins and highly glycosylated [46]. It reacts with Ref-1 and STAT3, and involved in the formation of STAT3–DNA complexes [47]. Although, ERp57 was reported to be up-regulated in cervical [47], liver, breast, rectal, thyroid and gastric cancer [48]. It was significantly down-regulated in the most of metastases and primary gastric cancers [49]. Similarly, in this study, ERp57 was down-regulated in *cis*-resistant cell line as compared to the sensitive counterpart cell line. Thus, the result together with literature data, indicate that dysregulation of the ERp57 expression enhances drug resistance and cancer progression while its level of expression is highly dependent on cell and cancer type. Hence, ERp57 may serve as a beneficial diagnostic marker and therapeutic target. Treatment of A2780^cisR^ cell line with synergistic combination of (CS and OA using 0/0 h) over-restored ERp57 expression while (CS and ART using 0/0 h) partially restored its expression, indicating a causal relationship between ERp57 expression and platinum resistance in ovarian cancer.

*PGAM1****,*** also known as PGAMA [15], is a glycolytic enzyme that converts 3-PG to 2-PG [47]. PGAM1 was over-expressed in different types of cancer such as hepatocellular carcinoma, breast carcinoma and colorectal cancer [50]. In the present study, PGAM1was not detected in A2780^cisR^ cells, possibly due to very low concentration. Given the important role of PGAM1 in the energy production associated with cell growth and cell proliferations, suggesting that PGAM1 may possibly be serving as a potential biomarker and therapeutic means towards Pt-resistance in ovarian cancer. Treatment with synergistic combination of (CS and ART using 4/0 h) over-restored PGAM1 expression, while treatment with synergistic combinations of (CS and ART using 0/0 h) partially restored PGAM1 expression. Whereas, after treatment with synergistic combinations of (CS and OA using 0/0 h) and (CS and OA using 0/4 h) PGAM1 expression was below threshold of detection. The results imply that level of PGAM1 expression was extremely sensitive towards the treatments that inhibited its ability to produce energy to nourish cancer cells thereby increased their vulnerability to the platinum drug. Hence, restoration of PGAM1 can be a contributing factor responsible for apoptosis enhancement so that the level of its expression can be a means for drug resistance in ovarian cancer. Therefore, chosen combinations can be helpful in sensitizing resistant cells towards platinum drugs and decreasing their side effects.

*LDHB,* also known as LDH-H [15], is involved in conversion of lactate to pyruvate [51]. LDHB was over-expressed in archival metastatic melanoma [52] and nasopharyngeal carcinoma [53]. However, it was found to be down-regulated in the MHCC97-H hepatocellular carcinoma cell strain against less metastatic MHCC97-L cell strain [54], and in the highly metastatic gallbladder carcinoma (GBC-SD18H) versus (GBC-SD18L) less potential metastasis cell line [55]. Similarly, in this study LDHB was down-regulated in the A2780^cisR^ cell line as compared to sensitive counterpart cell line. The findings indicate that aberrant LDHB expression is cell, stage and cancer type dependent, hence, targeting LHDB expression level would be beneficial diagnostic and therapeutic strategies. LDHB expression was partially/over-restored after treatment with the chosen combinations, supporting the idea that synergistic outcomes, besides other reasons are associated with enhancement of CS cytotoxic effects by ART and OA through LHDB restoration which have modulated its role in energy metabolism, suggesting a strong correlation between LDHB expression and apoptosis improvement.

*ATPA*, also known as ATP5A1, is a mitochondrial enzyme involved in the synthesis of ATP from ADP [15]. Over-expression of ATPA was observed in breast cancer cell (MCF7) [56], anaplastic thyroid cancer [57], acute lymphoblastic leukaemia [58], multidrug resistant cervical carcinoma (MDR) KB-v1 cells [59], and resistant leukaemia cells [60]. However, ATPA was down regulated in clear cell renal cell carcinoma [61], and chromophobe renal cell carcinoma [62]. In this study, ATPA was not detectable in A2780^cisR^ cell line as compared to its sensitive counterpart, possibly due to low concentration. The findings suggest that the dysregulation of ATPA level contributes to carcinogenesis event and targeting this protein might be of therapeutic importance. Treatment with synergistic combinations of (CS and OA using 0/4 h) and (CS and ART using 4/0 h) over restored its expression. ATPA expression after treatment with synergistic combinations of (CS and ART using 0/0 h) and (CS and OA using 0/0 h) was not detectable, suggesting that the nominated combinations are extremely effective in overcoming drug resistance in ovarian cancer model.

*ATP5H* “generates ATP from ADP in the existence of a proton gradient throughout the membrane.” [15]. ATP5H is overexpressed in lung adenocarcinomas [63] and MCF7 cell line [56]. Consistent with the reported findings, in the current study, ATP5H expression was up-regulated in A2780^cisR^ cell line as compared to the level found in sensitive A2780 cell line. The data point to the contribution of ATP5H in cancer formation and suggest that ATP5H may serve as a possible diagnostic marker and molecular target. ATP5H expression after the treatment with synergistic combinations of (CS and OA using 0/4 h), (CS and ART using 4/0 h), (CS and OA using 0/0 h), and (CS and ART using 0/0 h) was not detectable, suggesting that ATP5H expression is very sensitive to the combinations. Hence, these combinations would in helpful in opposing resistance in ovarian cancer.

*TPI* also known as TIM [15], is involved in the pathway of energy metabolism through catalysing the interconversion of DHAP to GAP [64]. TPI expression level was over-expressed in bladder squamous cell [61], prostate, lymph node, kidney, skin, testis, stomach, brain [65], epithelial ovarian cancer paclitaxel resistant A2780TC1 cells versus its sensitive cell line [66], and hepatocellular carcinoma versus normal tissues [50]. This is consistent with the current study wherein TPI was up-regulated in ovarian cancer cis-resistant versus its sensitive cell line. The findings suggest that dysregulation of TPI expression is playing a role in cancer promotion and drug resistance. Thus, it can be said that TPI can serve as potential biomarker and therapeutic targets. Treatment of A2780 cell line with synergistic combination of (CS and ART using 0/0 h) over-restored TPI expression. While treatment of A2780^cisR^ cell line with synergistic combination of (CS and ART using 4/0 h) partially restored its expression. Whereas, treatment with synergistic combination of (CS and OA using 0/0 h) fully-restored its expression, suggesting a strong correlation between TPI expression and improved cell kill.

*GOT1*, alternatively named as AATC [15], catalyses the interconversion of α-ketoglutarate and aspartate to glutamate and oxaloacetate [67]. Yu et al. reported that the elevation of GOT1 expression in pancreatic cancer was associated with shorter overall survival rate [68]. Similarly, Chakrabarti, G. reported that the elevated level of GOT1 in NSCLC resulted in poor outcome after radiotherapy was given, suggesting its involvement in the development of radio-resistance [69]. The same, in this study, GOT1 was up-regulated in cis-resistant cell line as compared to its sensitive A2780 cell line using A2780^cisR^ cell line as reference, suggesting GOT1 involvement in the development of drug resistance thus, it can be promising therapeutic and diagnostic tool. Treatment with synergistic combinations of (CS and ART using 0/0 h) partially-restored GOT1 expression. After treatment with synergistic combinations of (CS and OA using 0/0 h), (CS and OA using 0/4 h) and (CS and ART using 4/0 h), GOT1 expression was not detectable, suggesting that GOT1 is highly sensitive towards the treatments. Thus, it may play a major role in the synergistic outcomes, implying a causal relationship between GOT1 expression and drug resistance. 

*NM23*, relatively named as NDKA [15], is considered as metastasis-suppressor [70]. It is involved in large numbers of different biological activities [71], including cell migration, growth control, differentiation and signal transduction [67]. NM23 level was down-regulated in metastasis ovarian carcinoma [72] and nasopharyngeal carcinoma [73]. Likewise, in this study NM23 was down-regulated in A2780^cisR^ cell line as compared to the level found in sensitive A2780 cell line using A2780 cell line as reference, suggesting that down regulation of NM23 plays a role in apoptosis inhibition, pointing out that NM23 expression can be a prognostic marker and therapeutic target. Treatment with synergistic combinations of (CS and ART using 0/0) partially-restored NM23 expression. NM23 expression was not detectable after the treatment with the synergistic combinations of (CS and OA using 0/0 h), (CS and OA using 0/4 h) and (CS and ART using 4/0 h), suggesting that its expression is strongly sensitive towards the treatments. Thus, the combinations may be beneficial in overcoming drug resistance involving abnormal NM23 expression.

*IMMT*, also known as Mic60 [74], promotes protein import via (MIA) pathway and regulates cristae morphology and crista junctions [74]. It was down-regulated in prostate cancer androgen-independent (DU145) versus androgen-dependent (LNCaP) cell lines [75]. Similarly, in this study, IMMT expression was decreased in *cis*-resistant cell line as compared to its sensitive cell line, suggesting its usefulness as biomarker for platinum resistance in ovarian cancer. Treatment with synergistic combinations of (CS and ART using 0/0 h) and (CS and OA using 0/0 h) partially restored IMMT expression, suggesting that IMMT may be is playing a role in those synergistic outcomes, given that with the elevation of IMMT expression cell death was increased. Generally, this could be partly attributed to OA and ART actions which adjusted the molecular pathways including increasing Bax, decreasing Bcl-2, up-regulating of the caspases, and down-regulating NF-κB pathway thereby increased cell sensitivity towards platinum compounds leading to a greater growth inhibition. Therefore, the selected combinations can be a helpful in overcoming drug resistance involved abnormal IMMT expression.

*KPYM*, also known as PKM2 and PKM [15], catalyses the conversion of phosphoenolpyruvate to pyruvate [76]. It is a member of PK family [77], that plays an essential role in tumour growth and metabolism process [74], through its aerobic glycolysis ability [78,79]. KPYM was down-regulated in gastric carcinoma in cis- resistant cell line versus its sensitive cell line [76] and in colorectal cancer OX-resistant cell line versus its sensitive cell line. However, up-regulation of KPYM was reported in other type of cancers, including multiple myeloma, renal cell carcinoma, pancreatic cancer, ovarian, lung [77], breast, and colon cancer [80,81]. Likewise, KPYM in this study, was up-regulated in cis- resistant cell line versus its sensitive cell line using A2780^cisR^ cell line as reference. The results indicate that KPYM expression is cell and cancer type dependent and its expression is potential diagnostic and therapeutic targets. Treatment with synergistic combinations of (CS and ART using 0/0 h) and (CS and OA using 0/0 h) partially-restored KPYM expression. Whereas, after treatment with synergistic combinations of (CS and OA using 0/4 h) and (CS and ART using 4/0 h) KPYM expression was not detectable, implying that KPYM is very sensitive to the indicated combinations, suggesting a strong connection between enhanced apoptosis and KPYM expression. 

*PSAT*, also known and SERC [15], catalyses the formation of 3-phosphoserine from 3 phosphohydroxypyruvate [82], besides its involvement in serine biosynthesis [83]. The elevated expression of PSAT was observed in some cancer types, for example, in colon cancer [83] and clear cell ovarian carcinoma [84]. However, PSAT was not detectable in A2780^cisR^ cell line as compared to its parental A2780 cell line; this could be attributed to the detection issue or low concentration. After treatment, synergistic combinations of CS and ART (using 4/0 h) and CS and OA (using 0/0 h), PSAT expression was over-restored, suggesting that PSAT may be is participating in the final outcomes where the apoptosis initiation may perhaps be related to improvement of PSAT expression. Thus, the present data may point out PSAT as potential diagnostic and therapeutic targets.

*VDAC1*, also known as porin 31HM [15], regulates metabolic homeostasis and cell energy [85]. It interacts with antiapoptotic regulators (hexokinase, Bcl-Xl and Bcl-2). It is involved in releasing apoptotic factors presented in the mitochondria [85]. High level of VDAC1 was observed in various cancers, including cervical, lung, pancreatic, ovarian and thyroid cancers [86]. Likewise, in this study, VDAC1 expression was up-regulated in cis-resistant cell line as compared to parental cell line using A2780^cisR^ as a reference. The findings suggest that the VDAC1 is involved in tumour progression and that it may serve as useful biomarker and therapeutic target. After treatment with synergistic combinations of (CS and ART 4/0 h), (CS and OA using 0/4 h), (CS and ART 0/0 h) and (CS and OA using 0/0 h) VDAC1 expression was not detectable, suggesting that VDAC1 expression is significantly sensitive to the treatments. The result suggests that the rise in apoptosis could be attributed to a significant down-regulation of VDAC1 expression, suggesting a direct association between them, indicating the effectiveness of the selected treatments in restoring VDAC1 expression towards greater cell kill.

*ENOA* also known as Enolase 1 [15], is a metabolic enzyme that is implicated in pyruvate synthesis [87]. It facilitates the stimulation of extracellular matrix degradation and plasmin acting as a plasminogen receptor to promote tumour metastasis [87,88]. ENOA was elevated in in several malignancies, including breast, brain, gastric, cervix, colon, and kidney [84]. However, in the present study, ENOA was not detectable in the A2780^cisR^ cell line as compared to its sensitive cell line A2780. Similarly, four previous findings from the host laboratory reported the same finding [24,28,29,30], implying that ENOA is playing a significant role in development of platinum resistance in ovarian cancer. ENOA expression was over/partially-restored due to treatment with the selected combinations. It can thus be suggested that the treatments have inhibited ENOA ability to stimulate extracellular matrix degradation that has enhanced apoptosis induction. In general, therefore, it seems that the enhanced apoptosis could be to some extent ascribed to OA and ART pro-apoptotic ability to down-regulate NF-κB and Bcl-2 together with their cellular regulatory functions, which in turn has improved cancer cells sensitivity to CS, thereby enhanced apoptosis induction. 

*VIME* is class 3 of intermediate filaments family [15,89]. It is associated with tumorigenesis, progression and initiation of cancer [90]. VIME is up-regulated in different type of cancers such as lung, breast, prostate and colorectal [90]. Consistent with the reported findings, VIME was up-regulated in cisplatin-resistant A2780^cisR^ cell line as compared to its sensitive cell line A2780. The results indicate that VIME is playing a major role in cancer progression and reveal that VIME expression can be a valuable marker for various cancers. Treatment with synergistic combinations of (CS and OA using 0/4 h) over-restored its expression in the A2780 cell line. Likewise, treatment with synergistic combinations of (CS and ART using 0/0 h) and (CS and OA using 0/0 h) over-restored the VIME expression. It can thus be suggested that the synergism and observed improve in apoptosis is directly dependent on VIME manifestation. Where the treatments were effective in preventing its cell protection role, given that VIME provides resistance against stress and maintains cellular integrity [89]. Therefore, the nominated combinations can serve as useful means to sensitize cancer cells towards CS therapy.

*CAH2* is involved in extracellular acidification [91]. CAH2 was up-regulated in pancreatic and nervous system tumours [92], gastric carcinomas, and brain tumours [93]. In contrast, down regulation of CAH2 was reported in colorectal tumours [92] and hepatocellular carcinoma [93]. In this study, CAH2 was not expressed in A2780^cisR^ cell line as compared to its sensitive counterpart using A2780 as a reference, which may be due to low concentration or detection problems. Altogether, the findings suggest that the aberrant expression of CAH2 plays a role in cancer formation, metastasis, and resistance to chemotherapy; thus, it can serve as a promising diagnostic biomarker and therapeutic target. CAH2 expression was extremely sensitive to the treatment with synergistic combinations of (CS and ART using 4/0 h), (CS and OA using 0/4), (CS and ART using 0/0 h) and (CS and OA using 0/0), which might have modulated its acidification regulation activity in response to hypoxic environment leading to decrease cell proliferation that ultimately has increased cell death. Therefore, the results suggest that the restoration of CAH2 expression can be one of the causal factors for apoptosis induction. Thus, chosen combinations can be effective in combating drug resistance in ovarian cancer involving abnormal CAH2 expression.

*VINC* is located in cell-adherence junctions and in focal adhesions [94]. It influences adhesion protein turnover and contractility as well as regulates the cell signalling processes [94]. Vinculin may act as a metastasis inhibitor by decreasing cell motility and tumour inhibitor by assisting anchorage-dependent cell growth [94]. In this study, VINC was up-regulated in the cisplatin-resistant A2780^cisR^ cell line as compared to its sensitive counterpart using A2780^cisR^ cell line a reference, the result suggests the VINC expression as potential therapeutic and diagnostic approaches. Treatment with synergistic combinations of (CS and ART using 0/0 h) and (CS and OA using 0/0 h) partially-restored its expression in A2780^cisR^ cell line. Whereas, after treatment with synergistic combinations of (CS and ART using 4/0 h) and (CS and OA using 0/4 h) the expression of VINC was not detectable because of the effect of treatments, which by restoring its expression, may be altered protein turnover towards a greater protein degradation thereby enhanced growth inhibition. Hence, the result suggests that the significant reduction in VINC expression can be a causative factor accountable for improved growth inhibition.

*GBLP*, also known as RACK1 [15], is a cytosolic protein [92,93], which can interact with STAT, tyrosine kinases/phosphatases, PDE4D5, and PKC pathways, involved in cell movement, growth, division and adhesion [95]. GBLP was over-expressed in non-small cell lung cancer, melanoma and hepatocellular carcinoma [95]. In the present study, GBLP was below threshold of detection in the A2780^cisR^ cell line as compared to sensitive A2780 cell line. GBLP expression was over-restored due to the treatment with the selected combinations, suggesting that GBLP restoration is playing a contributory role in enhanced apoptosis resulting from chosen combinations which might have adjusted its behaviour thereby hindered the cell movement cell division towards a batter growth inhibition. It is likely that the significant improve in GBLP expression is directly associated with increased cells receptivity, suggesting GBLP as valuable diagnostic and therapeutic target.

*p18*, also known as COF1, is a member of COF family [15]. It controls actin dynamics involved in micropinocytosis, chemotactic movement, cell migration, phagocytosis, and proliferation [96]. Over-expression of p18 was reported in radioresistant astrocytomas [97,98], pancreatic cancer versus non-cancerous tissues [96], In this study, however, p18 was down-regulated in cis-resistant cell line as compared to its sensitive counterpart using A2780 as a reference. Although, the result is inconsistent with the reported findings, it can be said that this maker would be promising diagnostic and cancer therapeutic target given its important role in cell movement and propagation. After treatment with synergistic combinations of (CS and ART using 4/0 h) and (CS and OA using 0/4 h) (CS and ART using 0/0h) partially-restored p18 expression, suggesting that the observed increase in apoptosis level could be attributed to partial restoration of p18 expression. This may imply that p18 is possibly be involved in development of platinum resistance in ovarian cancer, taken into consideration its vital role as a member of cofilin family. 

*EIF5A1*, also known as IF5A1 [15], is one isoform of hypusinated eIF5A. It is an essential protein that undergoes many PTM, including acetylation and hypusination [98], which is involved in cell proliferation and the translation elongation stage [99]. EIF5A1 up-regulated in several malignancies, including colorectal carcinoma, lung adenocarcinoma, and glioblastoma [100]; for that reason, EIF5A1 is considered a tumour promoter (oncogene). In this study, EIF5A1 was down-regulated in cis-resistant cell line as compared to the level found in sensitive A2780 cell line. In support of this, Scuoppo et al. reported down-regulation of EIF5A1 in lymphoma [101] and knocking down of EIF5A1 has promoted the tumorigeneses of lymphoma, suggesting it as a tumour suppressor [101]. The findings suggest that EIF5A1 is playing a role in apoptosis inhibition and that its expression is cancer- and cell-type dependent. After treatment, the A2780^cisR^ cell line with synergistic combinations of CS and ART (using 0/0 h) and CS and OA (using 0/0 h) EIF5A1 expression was partially-restored, suggesting that the treatments were able to interrupt its role in protein biosynthesis, thereby prevented translation leading to cell death. Therefore, the results suggest a causal relationship between EIF5A1 expression and platinum resistance in ovarian cancer. Hence, the selected combinations can serve as useful treatments to sensitize cancer cells towards platinum therapy. 

*EF2* is one member of the elongation group, involved in protein translocation in the elongation stage [98]. EF2 was over-expressed in colorectal and gastric cancers [102], and that was suggested to be due to EF2’s ability to stimulate the cell cycle progression at G2/M, cdc2, and Akt, resulting in cell growth enhancement [103]. The result of the present study agrees with the above finding, EF2 was over-expressed in the A2780^cisR^ cell line, as compared to the A2780 cell line. The findings, together with the important role of EF2 in protein biosynthesis, clearly support the idea that elevation of EF2 expression is possibly be involved in ovarian cancer progression and drug resistance. Hence, the results represent EF2 as a new promising diagnosis biomarker and therapeutic target. Treatment, with synergistic combinations of CS and OA (using 0/4 h) and CS and ART (using 4/0 h) partially-restored EF2 expression. After the treatment with synergistic combination of CS and ART (using 0/0 h) EF2 expression was over-restored. This suggests that the observed rise in apoptosis degree is attributed to improvement of the CS cytotoxic effects by ART and OA actions towards opposing the drug resistance in ovarian cancer, indicating a causal relationship between EF2 expression and platinum resistance in ovarian cancer.

*EFTU* is another member of the elongation proteins family [15]. It regulates the phosphatidylinositol 4,5 bisphosphate and phosphatidylinositol 4-phosphate levels [100]. EFTU was reported to be up-regulated in the stomach, oesophageal, pancreatic, lung, and renal tumours [104]. In this study, EFTU was not expressed in the A2780^cisR^ cell line, as compared to its sensitive counterpart, using A2780, was used as a reference, and, consistent with this, Srisomsap et al. reported that EFTU was not expressed in the HepG2 cell line [105] Altogether, the findings indicate that EFTU expression is cancer- and cell type-dependent, suggesting that the EFTU expression can be a potential biomarker and therapeutic target. Treatment with synergistic combinations of CS and OA (using 0/0 h), partially-restored EFTU expression level, while treatment with synergistic combination of CS and ART (using 0/0 h) over-restored its expression, indicating that the treatment could be an effective way to sensitize cancer calls towards platinum compounds. 

*EF1G* is another member of the elongation proteins. Over-expression of EF1G was observed in gastrointestinal tract malignancies [106], acute myelogenous leukaemia [107], prostate, stomach, colon, lung, and breast cancer [108]. However, it was down-regulated in pancreatic cancers [109]. In this study, EF1G was not detectable (below threshold of detection) in A2780^cisR^ cell line as compared to the level found in the sensitive A2780 cell line. Taken together, the balance of evidence suggests that the EF1G expression is possibly cancer- or/and cell type -dependent, suggesting that targeting EF1G might be of therapeutic importance. Treatment with synergistic combinations of CS and ART 9using 0/0 h) and CS and ART (4/0 h) partially-restored EF1G expression, whereas, synergistic combination of CS and OA (using 0/4 h) over-restored its expression, suggesting that the restoration of EF1G by the treatments might have inhibited transformation by preventing translation of proteins during protein biosynthesis that have a direct role on cell growth reduction. Hence, EF1G expression may be is directly participating in the synergistic outcomes resulting from the combination of CS with OA and ART, where the improved apoptosis could be directly linked to EF1G restoration. Thus, selected combinations are effective in improving cancer cell sensitivity towards platinum drugs involved EF1G irregular expression.

*EF1A1* is the other member of this group, involved in protein synthesis during elongation stage [110], which is also known as EEF1A1 [15]. EF1A1 is associated with binding of G-actin, binding of microtubules, bundling of F-actin, and transporting β-actin mRNA [110]. It is over-expressed in pancreatic melanoma, lung, breast, colon, and prostate tumours [111]. Consistent with the reported findings, in the present study EF1A1 was up-regulated in A2780^cisR^ cell line as compared to its sensitive counterpart A2780. The findings suggest that EF1A1 plays an important role in apoptosis inhibition and development of drug resistance. After treatment with synergistic combinations of CS and OA (using 0/4 h) and CS and ART (using 0/0 h) EF1A1 expression was over-restored. While treatment with synergistic combination of (CS and OA using 0/0 h) partially-restored its expression, suggesting a straightforward connection between the elevated growth inhibition and decreased EF1A1 expression. This clearly supports that targeting EF1A1 can be a promising means to increase cancer cells sensitivity towards platinum drugs. Hence, selected treatments can serve as alternative approaches to overcome Pt-resistance through improving EF1A1 expression.

*hnRNPA1*, belong to mRNA processing proteins, which plays fundamental functions in gene expression regulation at translational and transcriptional levels [111], also are implicated in cell signalling, telomere biogenesis, and DNA repair [111]. *hnRNPA1*, also known as ROA1 [15], is up-regulated in a large number of cancers such as colorectal, breast, gliomas, and lung [112]. Similarly, in this study, hnRNPA1 was up-regulated in A2780^cisR^ cell line as compared to the level found in A2780 cells when A2780 was used as a reference, indicating that hnRNPA1 can be useful cancer biomarker and therapeutic target. Treatment of with synergistic combinations of CS and ART (using 0/0) and CS and OA (using 0/0) over-restored its expression. While after treatment with combinations of CS and ART (using 4/0) and CS and OA (using 0/4) hnRNPA1 expression was not detectable. Hence, the result suggests that the selected combinations have affected hnRNPA1 actions in gene expression and DNA repair, thereby increased cells vulnerability to CS cytotoxic action, leading to a better cell kill.

*hnRNPA2/B1*, also known as ROA2 [15], is another protein belongs to this group. It involves in various functions, including the regulation of mRNA metabolism, translation and transcription [113]. Up-regulated in serval cancer types such as pancreatic, lung [114], and breast [115]. Likewise, hnRNPA2/B1 was up-regulated in the cis-resistant cell line as compared to the level found in sensitive A2780 cell line. Altogether, the findings indicate its therapeutic value as diagnostic and medicinal target. It is worth noting that, in this study, hnRNPA2/B1 was named by MS for three spots (16 and 76 when A2780 was used as reference spot 18 when was A2780^cisR^ when used as a reference). After treatment with selected combinations, hnRNPA2/B1 expression was partially and over-restored, suggesting that the restoration of hnRNPA2/B1 expression possibly has altered its translation and transcription action, and, thereby, increased cell death. Hence, the result supports the possibility of a causal relationship between hnRNPA2/B1 expression and the resistance in ovarian caused by CS treatment.

*PGK1* is participating in the glycolytic pathway and contributing in angiogenesis events [116]. PGK1 was up-regulated in various cancers, such as multi-drug resistant ovarian cancer, breast cancer, renal cancer, pancreatic carcinoma, and squamous cell carcinoma [117,118,119]. Similarly, in the present study, PGK1 was up-regulated in A2780^cisR^ cells as compared to the level found in sensitive A2780 cell line when A2780^cisR^ was used as a reference. Thus, it can be said that PGK1 and its signalling genes can serve as future prognostic markers. Treatment with synergic combinations of CS and ART (using 0/0h), CS and ART (using 4/0h), and CS and OA (using 0/4 h) partially-restored PGK1 expression, suggesting the treatments might have influenced PGK1 angiogenic function, thereby preventing nutrients and oxygen, leading to a greater cell kill. However, its expression was further up-regulated due to the treatment with synergistic combination of CS and OA (using 0/0 h). From the above findings, it remains difficult to clearly establish clear association between PGK1 expression and enhanced cell kill and synergistic outcome. Hence, further studies would be necessary to gain better insight.

*PRDX1*, also known as PAG [15], is an antioxidant [120]. Its function interchanges from (peroxidase to a chaperone) with a high level of hydrogen peroxide [120]. Up-regulated level of PRDX1 was found in lung ovarian, liver, breast, gallbladder, prostate, and bladder cancer [120]. In this study, PRDX1 was below threshold of detection in cisplatin-resistant A2780^cisR^ cells as compared to the level found in A2780 cells when A2780 was used as a reference. However, Ren, Ye et al. reported a “relatively weak expression” of PRX1 in laryngeal cancer (Hep2), liver cancer cell (SUN449) lymphoma leukaemia (KOPN63) and acute lymphoblastic leukaemia (MOLT-4) cell lines [117]. The discrepancies, a result of unknown effect of PRDX1 or/ and its interchangeable role between peroxidase and chaperone functions. However, it can be said that PRX1 expression is cell, stage and cancer type dependent, suggesting a potential role of PRDX1 as therapeutic target and useful biomarker. Treatment with synergistic combinations of CS and OA (using 0/4 h) and CS and ART (using 4/0 h) partially-restored PRDX1 expression, while treatment with synergistic combinations of CS and OA (using 0/0 h) and CS and OA (using 0/0 h) over-restored PRDX1 expression, suggesting that the chosen combinations have repressed its antioxidant effect and enhanced its chaperone action towards a greater rate of growth inhibition. Thus, the result suggests a direct association between PRDX1 expression and apoptosis induction. Hence, the nominated combinations are beneficial in overcoming drug resistance in ovarian cancer involved abnormal expression of PRDX1. 

*PRDX6*, also known as 1-Cys PRX, is a bifunctional enzyme [121] where it has a phospholipase A2 and glutathione peroxidase activities [121]. It is involved in tumour metastasis, proliferation and protection [121], playing both anti cytoprotective or cytoprotective roles [122]. PRDX6 was up-regulated in various cancers such as breast [121], lung [122], and oesophageal cancer [64]. In the present study, PRDX6 was below threshold of detection, in the A2780^cisR^ cell line as compared to the level found in A2780 cells when A2780 was used as a reference, however, RDX6 was very sensitive to selected treatments, suggesting a direct association between promotion of apoptosis and PRDX6 level of expression. Given its bifunctionality, it is possible that targeting PRDX6 might be of therapeutic importance. 

*Op18*, also known as STMN1 [15], is a highly conserved phosphoprotein involved in cell proliferation, morphogenesis, and differentiation [123]. It plays a role in the regulation of the microtubule filament system through microtubules destabilization [15]. Fang et al. reported that Op18 was over-expressed in different cancers, including osteosarcoma, lung, breast, and bladder [124]. In contrast, in this study, Op18 was down-regulated in cisplatin-resistant A2780^cisR^ cells as compared to its sensitive counterpart using A2780 as a reference. However, the result is in line with previous findings from the host laboratory using the same cell lines [24,28], suggesting that Op18 expression is a cell- and cancer type-dependent. Despite these discrepancies, it can be said that Op18 can serve as a potential biomarker for the resistance in ovarian cancer. Treatment with synergistic combination of (CS and ART using 0/0 h) over-restored Op18 expression, suggesting the selected treatment may have disturbed its morphogenetic signalling preventing cell proliferation that in turn led to growth inhibition, indicating that treatment could be an effective means to sensitize cancer calls towards cisplatin.

*MCM7*, also known as CDC47 [15], is a member of the MCM complex [125] involved in the regulation of cell cycle progression and licensing DNA replication [126]. MCM7 was over-expressed in different human cancers, including neuroblastoma, colorectal and prostate [127]. In this study, MCM7 was not detected in A2780^cisR^ cell line as compared to the level found in sensitive A2780 cell line. Treatment with the synergistic combinations of CS and OA (using 0/0 h) over-expressed MCM7 expression. While treatment with the combination of CS and ART (using 0/0 h) fully-restored its expression. This could be partially attributed to ART inhibitory ability of the important cell cycle regulators, including CDK4, CDK2 [13], cyclin A, cyclin E, cyclin D3, cyclin D1, CDK6, and E2F1 [17], given that MC7 is associated with cell cycle regulation, it is possible that ART was able to recover MCM7 expression as a one of the cell cycle regulators proteins, which in turn have improved CS cytotoxic action. Therefore, the results suggest a direct correlation between improved cell death and MCM7 expression and indicate that the selected combinations are valuable in opposing of Pt resistance in ovarian cancer.

*CLIC1*, also known as NCC27 [15] is one of CLIC family which involved in the regulation of trans-epithelial transport, cell volume, electrical excitability, pH levels, ion homeostasis, cell adhesion, cells apoptosis and cell cycle [55]. In the present study, CLIC1 was down-regulated in cisplatin-resistant cell line as compared to the level found in the sensitive A2780 cell line when the A2780^cisR^ cell line was used as a reference. Treatment with synergistic combination of CS and ART (using 0/0 h) over-restored its expression, whereas, treatment with synergistic combination of CS and OA (using 0/0 h) further down-regulated CLIC1 expression. These obviously confusing findings suggest that CLIC1 may possibly not be playing any important role in those outcomes. However, taken into consideration CLIC1′s roles in chloride channel [128], cell adhesion, cells apoptosis, and cell cycle [61], suggesting that targeting CLIC1 might be of therapeutic importance. Additional studies at a molecular level would be required to clarify the matter.

*RAN* is a member of the Ras super-family [125], also known as GTPase Ran [15]. It promotes the assembly of bipolar spindle in mitosis phase [129]. The elevated level of RAN was reported in different cancers such as ovarian, stomach, kidney, colon, and lung cancer [125]. Similarly, in this study, RAN was up-regulated in cis-resistant cell line as compared to its sensitive counterpart using A2780^cisR^ cell line as reference. Silencing of RAN inhibited cell growth, or/and induced apoptosis in colon, nasopharyngeal, renal breast, and ovarian tumours [128]. The findings suggest that RAN expression is a promising diagnostic marker and therapeutic target. Treatment with synergistic combination of CS and ART (using 4/0 h) over-restored RAN expression, whereas, treatment with synergistic combinations of CS and OA (using 0/4 h), CS and ART (using 0/0 h), and CS and OA (using 0/0 h) partially-restored its expression, suggesting the treatments have suppressed RAN activity of bipolar spindle assembly in mitosis phase leading to a greater growth inhibition. Therefore, the results indicate that the reduction in RAN expression can be directly accountable for the observed rise in cell kill. Hence, the selected treatments can be a potential therapeutic strategy to sensitize ovarian cancer cells towards Pt drugs.

Moreover, *1433Z* is another signal transduction protein, it is one isoform of the 14-3-3 family, which exhibits a pro-oncogenic role in multiple tumour types [130]. The 1433Z is the key regulator of the major cellular processes [131], including autophagy, progression, adhesion, differentiation, proliferation, and apoptosis [131]. The 1433Z was down-regulated in A2780^cisR^ cells as compared to its sensitive counterpart, when A2780 was used as reference. Likewise, previous findings reported down-regulation of 1433Z in (A2780-TR, TA) paclitaxel-resistant as compared to its sensitive counterpart [132], and in (SKOV3-TR, TS) as compared to its sensitive counterpart [132]. However, up-regulation of 1433Z expression was reported with others cancers such as pancreatic [131], head and neck, NSCLC, breast [127], hepatocellular carcinoma, and stomach cancer [133]. Overall, the results indicate that 1433Z expression is cancer and cell dependent, suggesting that the level of its expression is a potential diagnostic marker for Pt-drug resistance in ovarian cancer, Treatment with synergistic combinations of CS and ART (using 0/0 h) and CS and OA (using 0/0 h) over-restored 1433Z expression, while treatment with synergistic combination of (CS and OA using 0/4 h) partially-restored its expression, suggesting that the growth inhibition produced by the combination of OA and ART with CS, was caused by the down-regulation of Bcl-2 and NF-κB mediated by other mechanisms. For example, in the case of the combination with ART the final outcomes could be ascribed to the inhibition of cyclin A, cyclin E, cyclin D3, cyclin D1, CDK6, E2F1 and JAB1 transcriptions [17], CDK4, CDK2 expressions [13], S6K, PI3K, mTOR, Akt [10] and enhancement of IFIT3, p21, and p27 [17]. This has increased the influx of CS or/and decreased its efflux. Additionally, the selected combinations might have manipulated 1433Z effect on cell adhesion, cell differentiation, and cell proliferation, which in turn has prompted apoptosis induction. Thus, the results overall suggest that the selected combinations are beneficial for 1433Z restoration towards overcoming drug resistance in ovarian cancer.

Two subunits of proteasome PSA3and PSA7 were also found to be differentially expressed, are involved in the regulation of several crucial cellular process, including apoptosis, immune responses, DNA repair, protein quality control and cell cycle progression [134]. PSA3 (also known as PSMA3) [15], was down-regulated in cis-resistant cell line as compared to its sensitive counterpart when A2780 was used as a reference, this is consistent with finding of Moghanibashi et al. in which PSA3 was reported to be down-regulated in ESCC [135]. PSA7, also known as PSMA7 [15]. It was up-regulated in cis-resistant cell line as compared to its sensitive counterpart when A2780^cisR^ cell line was used as a reference. Up-regulation of PSA7 expression was reported previously in colorectal cancer [136]. The results suggest that PSA3 and PSA7 may serve as potential diagnostic and therapeutic targets. PSA3 and PSA7 expressions after treatment with selected combinations were either partially or over-restored, suggesting that the treatments may be have decreased PSA3 and PSA7 DNA repair ability and recovered their growth inhibition ability that have increased the cell cycle arrest and decreased protein quality control leading to a greater growth inhibition. Hence, the result suggests a causal relationship between PSA3 and PSA7 expressions and resistance to cisplatin therapy, signifying the effectiveness of the applied treatments in overcoming drug resistance involved abnormal PSA3 or/and PSA7 expression in ovarian cancer.

## 4. Materials and Methods 

CS, OA, ART, trypsin, HEPES, triton-X 100, PBS, MTT, DMSO, and NaOH (Sigma-Aldrich, Castle Hill, Australia), K_2_[PtCl_4_] (Sigma, USA), KI (BDH Chemicals, Queensland, Australia), A2780, A2780^cisR^ and A2780^ZD0473R^ ovarian cancer cell lines were kindly provided by Dr Philip Beale (NSW Cancer Centre, Royal Prince Alfred Hospital (RPAH), Sydney, Australia), Varian Cary 1E UV/VIS spectrophotometer, Varian SpectrAA240 atomic absorption (AAS) with GTA-120 graphite furnace tube atomizer at the host laboratory (School of Medical Sciences, The University of Sydney), 28% ammonia solution (UNIVAR, Downers Grove, IL, USA), AgNO_3_ (BDH Chemicals), FCS, 5× RPMI1640 media, L-glutamine, and NaHCO_3_ (5.6%) (Thermo Trace Pty Ltd. Melbourne, Australia), 25 cm^2^ cell culture flasks (Crown Scientific, San Antonio, TX, USA), carbon dioxide incubator (SANYO, Osaka, Japan), JETQUICK DNA Spin Kit (GENOMED GmbH, Löhne, Germany), Tris (hydroxymethyl) aminomethane hydrochloride (Sigma, Munich, Germany), CalcuSyn software version 2 (Bio-Rad, Cambridge, UK), urea and CHAPS (Calbiochem, Darmstadt, Germany), thiourea and DTT (Merck, Darmstadt, Germany). Melanie version 7.0 (SIB, GeneBio, Geneva, Switzerland). glycerol and SDS (MP Biomedicals, LLC, Illkirch Cedex, France), DeStreak reagent (GE Healthcare Bio-sciences AB, Uppsala, Sweden), criterion TGX Precast gel, iodoacetamide, carrier ampholytes, PROTEAN i12 IEF system, mineral oil, and 10 X tris/glycine buffer (Bio-Rad, Irvine, CA, USA), agarose, wicks, ChemiDoc XRS imaging system, Bio-Safe Coomassie Stain, Ready-Strip™ IPG Strip, criterion Dodeca cell, protein assay standard II kit, bromophenol blue, Microplate Reader Model 3550, and automated cell counter (Bio-Rad, Gladesville, Australia) 96 well flat bottom plates (Edward Keller), and KCl (BDH Chemicals).

### 4.1. Cell Culture and Subculture

Cell culturing was carried out following the same procedure as described previously [137]. Briefly, cells were grown in flasks 25 cm^2^, incubated in 95% air, 5% CO_2_ at 37 °C. Cells were preserved in (log) growth phase in RPMI 1640 containing heat-inactivated fetal calf serum (10%), bicarbonate (0.112%), HEPES (20 mM), and glutamine (2 mM). A2780 cell line was subjected to increasing intensity of CS until it gained resistant (A2780^cisR^). Moreover, A2780 cell line was treated continuously with increasing strengths of ZD0473 to develop resistant to the drug [136].

### 4.2. Drugs Preparation

Cisplatin was dissolved in dimethyl formamide (DMF) followed by addition of Milli-Q water (at the ratio of 1:5) to give 1 mM stock solution. Oleanolic acid (OA) was dissolved in ethanol to attain 1 mM concentration. Artemisinin (ART) was dissolved in ethanol and Milli-Q (mQ) water (at a ratio 50:50) to attain 1 mM concentration.

### 4.3. Single-Drug Treatments

Stock solutions were subjected to serial dilutions to give final concentrations ranging from 0.16 to 200 μM, using 10% RPMI1640 medium. Cells at a density of 4000 to 6000 cells/well were treated with CS, OA, and ART, individually at four different concentrations, repeated three times in the same plate, and incubated for 72 h. Two controls were used; one contained cell and medium and the other contained (cells, medium and ethanol). Cytotoxic effects of CS, OA, and ART alone at different concentrations were evaluated using MTT assay following the procedure described by Mosmann [137]. Briefly, after 72 h of incubation with drug(s), medium was replaced by adding fifty μL of MTT solution to each well followed by further 4 h incubation. Then one hundred fifty μL of DMSO was added to each well to dissolve formazan crystals after MTT solution was aspirated. The absorbance at 570 nm was measured using microplate reader [137]. Percentage of living cells was calculated by dividing the average of optical density of wells (containing treated cells) by the average of optical density of wells (containing untreated cells). Concentration of (CS, OA, or ART) required to produce 50% of growth inhibition in each cell line compared to control (which is referred to as IC_50_ value) was obtained from the standard curve, representing the percentage of the cell viability versus the concentration.

### 4.4. Combination Studies

Cells were treated with solutions of compounds alone and in combination at three different concentrations, generally at constant ratios of their IC_50_ values. The concentration ranges were: cisplatin: 0.13–2.11 μM, 1.29–20.61 μM and 1.67–26.78 μM; oleanolic acid: 6.80–108.80μM, 5.23–83.68 μM, and 2.17–34.72 μM; artemisinin 3.36–53.70 mM, 5.36–85.76 mM, and 7.27–116.35 mM for A2780 (parent), A2780^cisR^ (cisplatin resistant) and A2780^ZD0473R^ (ZD0473-resistant) cell lines respectively. Briefly, 100 μL of cells seeded in each well of 96-well plate were treated with CS, OA, and ART alone and in combination with CS using the chosen concentrations, following three sequential modes; (Pt/Phyt: 0,0 h), (Pt/Phyt: 0/4 h) and (Pt/Phyt: 4/0 h), non-treated cells served as control. The plates were incubated for 72 h with each experiment repeated three times. The inhibition of cell growth was determined using the (MTT) reduction assay as previously mentioned. The combination index (CI) value of two compounds in combination was determined according to the method developed by Chou and Talalay [14]. CI of <1, =1 and >1, indicates synergism, additiveness, and antagonism respectively [14,138]. 

### 4.5. Platinum Cellular Accumulation and Platinum DNA Binding Studies

A2780 and A2780^cisR^ cells with density of (5 × 10^6^ mL^−1^) in each petri dish treated with CS alone and its binary combinations with OA and ART using fixed concentrations of their IC_50_ values, were incubated for forty-eight hours. Cells were collected and centrifuged at 3500 rpm for two minutes at 4 °C. Next, cell pellets were washed twice with 4 mL of cold PBS and centrifuged at 3500 rpm at 4 °C for two minutes. Cell pellets were then re-suspended with 0.5 mL of cold PBS. Then (1.5 μL) was taken from each sample and mixed with the same quantity of trypan blue to determine the cell survival fraction using the automated cell counter. The samples were then re-spun for two minutes at 10,000 rpm at 4 °C then cell pellets were stored at −20 °C until assayed.

### 4.6. Cellular Accumulation

Cell pallets were lysed with 0.5 mL of 1% triton-X100 added to each sample and sonicated in ice for thirty minutes then centrifuged at 14,000 rpm for three minutes. Then 400 μL of the supernatant from each sample was used to determine cellular platinum levels by measuring the absorbance values using graphite furnace atomic absorption spectrometry. The platinum level in each sample was calculated as nmol of platinum per 5 × 10^6^ cells.

### 4.7. Platinum–DNA Binding

Cell pellet from each sample was re-suspended in 200 μL of PBS for DNA extraction using JETQUICK DNA Spin Kit following the protocol of the provider’s instructions (see ref, [139]. The DNA content was calculated from absorbance as A260 nm × 50 ng/μL [140]. A260/A280 ratio [140] between 1.75 and 1.8 for each sample ensured acceptable purity for tested samples. Next, 200 μL from each DNA sample was used for platinum level quantification using graphite furnace AAS.

### 4.8. Proteomic Studies

The proteomic studies were initially intended: (1) to determine the proteins that underwent changes in expression in A2780^cisR^ as compared to parent A2780 cell line and (2) to evaluate the effect of the selected combination treatments on those proteins. The studies were based on 2-D gel electrophoresis and the use of Melanie 7.0 software. The ultimate objective was to identify protein biomarkers that might be associated with the resistance towards CS, using MALDI TOF/TOF MS/MS. The identified spots were matched against Mascot (http://www.matrixscience.com), Swiss-Prot database (http://www.uniprot.org/), at APAF (http://www.proteome.org.au/). *“This work was undertaken at APAF the infrastructure provided by the Australian Government through the National Collaborative Research Infrastructure Strategy (NCRIS).”*

### 4.9. Sample Preparation, Treatments, Pellet Collection and Protein Quantification

A2780/ A2780^cisR^ (5 × 10^4^) cells were seeded and incubated for 24 h. The next day, A2780^cisR^ cells were treated with chosen drug combinations, namely CS and ART (using 4/0 h), CS and OA (using 0/4 h), CS and ART (using 0/0 h), and CS and OA (using 0/0 h); then all samples (including untreated samples from both cell lines) were incubated for 24 h. Cells then were collected, centrifuged (at 3500 rpm, 2 min at 4 °C), washed (PBS/5 mL), re-suspend (by PBS/1 mL), counted, and centrifuged (at 14000 rpm, 2 min at 4 °C). Following this, the pellets were lysed in 500 μL of: (501 µL) of 65 mM DTT, (24.04 g) of 8 M Urea, (2.00 g) of 4%CHAPS, (7.61 g) of 2 M thiourea, (5 tablets) protease inhibitor and (up to 50 mL) of mQ water. Subsequently, the pellets were lysed and centrifuged at (13, 000 rpm and 4 °C) for thirty minutes. Next, the supernatant was collected and stored at −80 °C (see refs [141]. Total protein was measured as described by the assay kit provider [142], based on Bradford method [143]. The absorbance was measured at 595 nm using microplate reader BIO-RAD Model 3550. The standard curve was used to determine protein concentration of the samples.

### 4.10. Two-Dimensional Gel Electrophoresis

For protein separation, first-dimension IEF was performed following the technique of the provider’s manual * [144]. These Bio-Rad manuals do not have author or published dates; therefore, (*, ** and ***) was placed next to the reference to distinguish each instruction manual. However, the URL of each one is provided at the reference list. Moreover, they can be found at the company’s website (https://www.bio-rad.com/) through their distinct bulletin numbers (bulletin #4110009, bulletin #2651, and bulletin #4006197). Briefly, the IPG strip (11 cm L× 3.3 mm W × 0.5 mm thick/3–10 NL pH gradient) was passively rehydrated in 180 μL of rehydration solution containing: 4% CHAPS 100 μL, 0.0002% bromophenol blue, 60 mM DTT, 8 M Urea, (15 mg/1 mL) DeStreak, 2 M thiourea, and 0.2% ampholytes with 200 µg of protein from each sample then the samples were kept overnight. Then IEF procedure was performed using PROTEAN i12 IEF system following the manufacturer’s running protocol ** [145]. The 2nd dimension was carried out using IPG strip, precast gels and criterion Dodeca cell unit, following the procedures described in the product’s instruction manual *** [146]. Briefly, IPG strips were incubated in (traces of bromophenol, 20% glycerol *v*/*v*, 2% DTT *w*/*v*, 50 mM Tris–HCl pH 8.8, 2% SDS *w*/*v*, and 6 M urea) then were kept on a rocker for fifteen minutes ** [145] Again, the strips were incubated in (traces of bromophenol, 2% glycerol *v*/*v*, 50 mM Tris–HCl pH 8.8, 2% SDS *w*/*v*, 2% IAA *w*/*v*, and M urea) solution and left on a rocker for fifteen minutes ** [145]. Next, IPG strips were washed with (192 mM glycine, 0.1% (*w*/*v*) SDS, pH 8.3, and 25 mM Tris). The Criterion Dodeca Cell was filled with 6 L of (6 X) Tris/Glycine/SDS (0.1% SDS, 192 mM glycine, and 25 mM tris, pH (8.3). Before running the program, gel cassettes were sealed with a melted agarose ** [145]. Sequentially, the unit was run for sixty-five minutes at 200 V *** [146]. The gels were dyed with Coomassie blue, scanned with the ChemiDoc XRS imaging system and saved as TIFF format (see, [147]. The paired gel images (12 gels, a duplicate for each gel sample) from untreated A2780 cell line, untreated A2780^cisR^ cell line, treated A2780^cisR^ cell with the following combinations: CS and ART (using 4/0 h), CS and OA (using 0/4 h) and CS and ART (using 0/0 h) and CS and OA (using 0/0 h), and were imported into Melanie 7.0 software and clustered into six groups for spots detection and quantification, following the product’s manual instructions (see, [148]. Briefly, the spot from the same sample was manually and automatically matched and paired using two landmarks. Next, the matched spots were annotated and assigned identical matched ID in all gel samples. Following this, the six groups were assembled into four hierarchical classes: (1) untreated A2780, (2) untreated A2780^cisR^, (3) treated A2780^cisR^ with (CS and OA), and (4) treated A2780^cisR^ with (CS and ART), subsequently, spots were automatically analysed and compared across all the samples using A2780 cell line as a reference after that untreated A2780^cisR^ cell line as reference then spots were quantified automatically. In this study, a two-fold or greater change in protein expression was considered significant. Based on this parameter, 101 spots out of 133 spots were selected and sent to APAF (Macquarie University) for protein identification using MALDI-TOF and Mass Spectroscopy through APAF (http://www.proteome.org.au/). The detailed methods of MALDI and MS were previously described [149,150,151]. Briefly, at APAF (Macquarie University) (http://www.proteome.org.au/), using 4800 plus MALDI TOF/TOF Analyser (AB Sciex), spots were excised (spot cutter/ Bio-Rad), de-stained (ammonium bicarbonate / acetonitrile) and digested (ammonium bicarbonate, 16 h at 37 °C). Next, the peptides were extracted (0.1% TFA/C18 zip-tip) and yield (MALDI-MS). The most intense (8) peptide peaks (ions) from each spot were further fragmented by (CID system). Masses and intensities were measured by MS/MS (TOF-TOF) and matched against Mascot database (Matrix Science Ltd., London UK. (http://www.matrixscience.com/). Monoisotopic peak lists were matched against Homo sapiens using Swiss-Prot database (http://www.uniprot.org/) [15]. The score was calculated according to Mascot parameter (http://www.matrixscience.com/) and as previously reported (greater than 56 is significant, *p* < 0.05).

## 5. Conclusions

The results show that synergistic effect from combinations of CS with OA and ART in A2780, A2780^cisR^, and A2780^ZD0473R^ are dependent on the dose, time, and cell type. The results show that the increased CS-DNA binding levels was consistent with the observed increase in CS-accumulation. The results also show that platinum-DNA levels are greater in the A2780^cisR^ cells, indicating that the presence of ART and OA have modulated some of the platinum resistance mechanisms, such as decreases of the platinum-DNA adducts formation [18], increases of platinum detoxification, and increases of DNA repair [19].

Investigation of proteins expression has shown that out of 101 proteins, (49) proteins were 2 ≥ fold differently expressed and were successfully identified by MS. Those proteins were shown to play a critical role in cell cycle regulation, pro-survival, and anti-survival pathways, suggesting their participation in platinum drug resistance. Most of the identified proteins are associated with molecular chaperones and stress, metabolism, and invasion and metastasis. It can, thus, be suggested that those identified proteins can be additional novel predictive factors for early detection to individualised therapeutic interventions, considering the differences of each histologic type to improve the overall survival rate along with preserving the quality of the patient life. Finally, this study suggests that these synergistic combinations were able to restore back most of the protein expressions in the A2780^cisR^ cell line; thus, they can be deemed as valuable approaches to circumvent Pt-resistance associated with aberrant expression of those proteins in ovarian cancer. Based on this study, evaluations of cellular sensitivity/resistance to those treatments in vivo studies using suitable animal models deserve to be further tested in the future. Studies on cellular DNA damage, oxidative stress, and cell cycles may provide further mechanistic information.

## Figures and Tables

**Figure 1 ijms-21-07500-f001:**
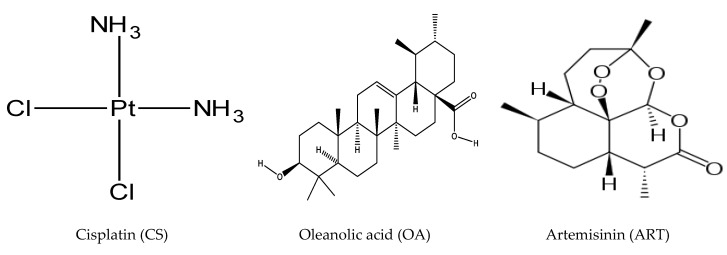
Chemical structure of cisplatin, oleanolic acid, and artemisinin.

**Figure 2 ijms-21-07500-f002:**
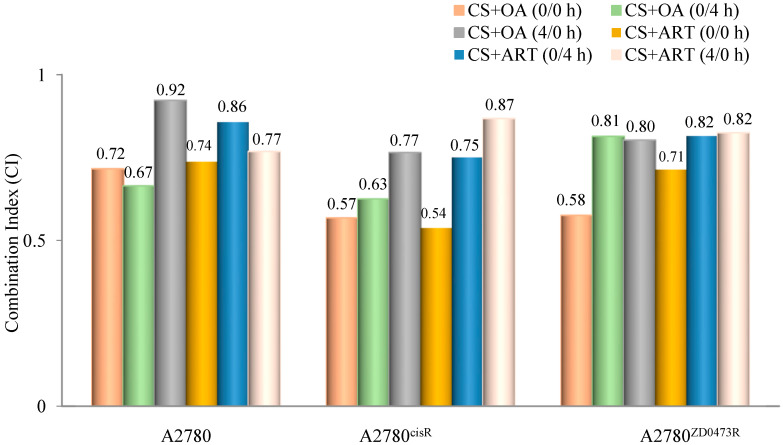
Graphical representation of combination indices (CI) applying to binary combinations of CS with OA and ART at ED50 (where drugs were added in equipotent ratios based on IC_50_ values).

**Figure 3 ijms-21-07500-f003:**
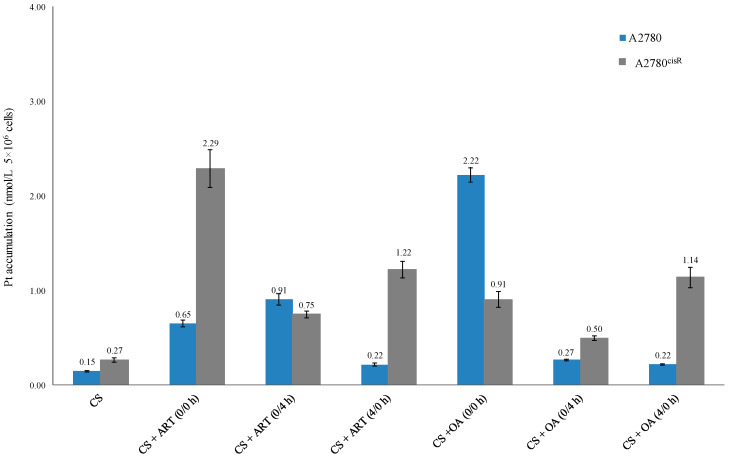
Platinum accumulation level from CS alone and in combination with ART and OA.

**Figure 4 ijms-21-07500-f004:**
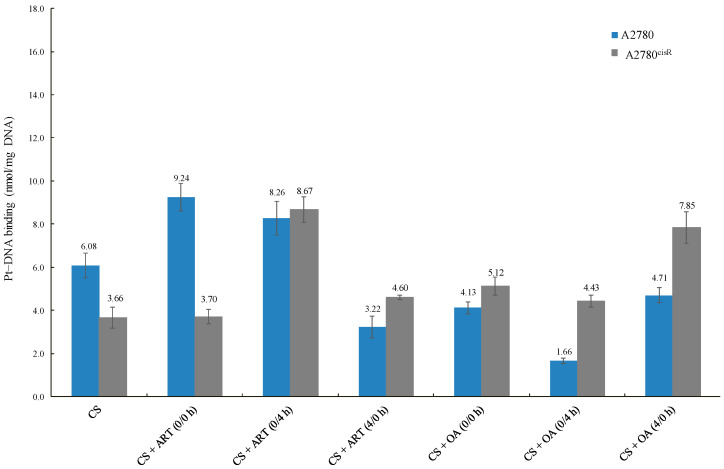
Platinum–DNA-binding levels of CS alone and in combinations with ART and OA.

**Table 1 ijms-21-07500-t001:** IC_50_ values of cisplatin (CS), artemisinin (ART), and oleanolic acid (OA) as applied to the human ovarian cancer cell lines, A2780 A2780^cisR^, and A2780^ZD0473R^ cell lines based on, at least, triplicate measurements, with the period of incubation being 72 h.

IC_50_ (μM) and RF Values					
Drug	A2780	A2780^cisR^	*RF	A2780^ZD0473R^	RF
CS	0.66 ± 0.08	6.44 ± 0.11	9.75.63	8.37± 0.06	12.6
ART	16.78 ± 0.06	26.8 ± 0.13	1.60	36.36 ± 0.09	2.16
OA	34.0 ± 0.31	26.15 ± 0.28	0.76	10.85 ± 0.05	0.3

*RF: resistant factors.

**Table 2 ijms-21-07500-t002:** Combination index (CI) values of CS with OA and ART in the A2780 cell line, where drugs were added in equipotent ratios based on IC_50_ values.

CI Value at								
Drug	Sequence (h)	Molar Ratio	ED_50_	ED_75_	ED_90_	D_m_	m	r
CS			N/A	N/A	N/A	1.16	0.88	0.99
OA			N/A	N/A	N/A	37.19	0.64	1.00
CS + OA	0/0	1:51.52	0.72	0.50	0.37	0.53	1.06	0.99
CS + OA	0/4		0.68	0.59	0.54	0.49	0.86	0.97
CS + OA	4/0		0.92	1.07	1.28	0.69	0.72	0.98
ART			N/A	N/A	N/A	26.5	0.55	0.99
CS + ART	0/0	1:25.42	0.74	0.55	0.47	0.40	0.85	1.00
CS + ART	0/4		0.86	1.05	1.46	0.47	0.62	0.99
CS + ART	4/0		0.77	0.55	0.60	0.32	0.72	0.99

Dm: medium effect dose; m: exponent defining shape of the dose effect curve; r: reliability coefficient.

**Table 3 ijms-21-07500-t003:** CI values of CS with OA and ART in the A2780^cisR^ cell line where drugs were added in equipotent ratios based on IC_50_ values.

			CI Values at					
Drug	Sequence (h)	Molar Ratio	ED_50_	ED_75_	ED_90_	Dm	m	r
CS			N/A	N/A	N/A	8.03	0.63	0.99
OA			N/A	N/A	N/A	56.44	0.40	1.00
CS + OA	0/0	1:4.05	0.57	0.49	0.50	2.88	0.61	1.00
CS + OA	0/4		0.63	0.42	0.32	3.15	0.72	1.00
CS + OA	4/0		0.77	0.53	0.43	3.85	0.70	0.99
ART			N/A	N/A	N/A	77.82	0.49	1.00
CS + ART	0/0	1:4.16	0.54	0.88	1.51	2.98	0.48	1.00
CS + ART	0/4		0.75	0.67	0.64	4.17	0.65	1.00
CS + ART	4/0		0.87	0.80	0.78	4.82	0.64	1.00

D_m_: medium effect dose; m: exponent defining shape of the dose effect curve; r: reliability coefficient.

**Table 4 ijms-21-07500-t004:** CI values of CS with OA and ART in the A2780^ZD0473R^ cell line where drugs were added in equipotent ratios based on IC_50_ values.

			CI Values at					
Drug	Sequence (h)	Molar Ratio	ED_50_	ED_75_	ED_90_	D_m_	m	r
CS			N/A	N/A	N/A	11.35	0.83	0.98
OA			N/A	N/A	N/A	29.43	0.43	0.96
CS + OA	0/0	1:1.3	0.58	0.61	0.86	3.55	0.61	0.99
CS + OA	0/4		0.81	0.56	0.52	5.00	0.80	0.99
CS + OA	4/0		0.80	0.71	0.74	5.53	0.73	0.99
ART			N/A	N/A	N/A	17.19	0.57	0.99
CS + ART	0/0	1:4.35	0.71	0.80	0.95	1.46	0.56	1.00
CS + ART	0/4		0.82	0.91	1.36	1.71	0.53	0.99
CS + ART	4/0		0.82	0.85	1.09	1.89	0.58	1.00

D_m_: medium effect dose, m: exponent defining shape of the dose effect curve, r: reliability coefficient.

**Table 5 ijms-21-07500-t005:** Platinum accumulation level of CS alone and in combination with ART and OA.

	* Platinum Accumulation			
	A2780		A2780^cisR^	
DRUG	Value	Fold Change	Value	Fold Change
CS	0.15 ± 0.01	-	0.27 ± 0.02	-
CS + ART (0/0 h)	0.65 ± 0.04	4.33	2.29 ± 0.20	8.48
CS + ART (0/4 h)	0.91 ± 0.06	6.06	0.75 ± 0.04	2.77
CS + ART (4/0 h)	0.22 ± 0.02	1.46	1.22 ± 0.09	4.51
CS + OA (0/0 h)	2.22 ± 0.08	14.8	0.91 ± 0.08	3.33
CS + OA (0/4 h)	0.27 ± 0.01	1.8	0.50 ± 0.02	1.85
CS + OA (4/0 h)	0.22 ± 0.01	1.46	1.14 ± 0.11	4.22

* Expressed as: nmol Pt per 5 × 10^6^ cells.

**Table 6 ijms-21-07500-t006:** Platinum–DNA-binding levels of CS alone and in combinations with ART and OA.

	* Platinum–DNA Binding Level			
	A2780		A2780^cisR^	
DRUG	Value	Fold Change	Value	Fold Change
CS	6.08 ± 0.57	-	3.66 ± 0.49	-
CS + ART (0/0 h)	9.24 ± 0.64	1.52	3.70 ± 0.33	1.01
CS + ART (0/4 h)	8.26 ± 0.78	1.36	8.67 ± 0.59	2.37
CS + ART (4/0 h)	3.22 ± 0.49	0.53	4.60 ± 0.11	1.26
CS + OA (0/0 h)	4.13 ± 0.28	0.68	5.12 ± 0.43	1.40
CS + OA (0/4 h)	1.66 ± 0.11	0.27	4.43 ± 0.29	1.21
CS + OA (4/0 h)	4.71 ± 0.36	0.77	7.85 ± 0.73	2.14

* Expressed as: nmol Pt per mg of DNA.

**Table 7 ijms-21-07500-t007:** List of protein expressions in A2780, compared to A2780^cisR^ using A2780 as reference.

Match ID	Expression in A2780^cisR^/A2780	CS + ART (4/0 h)	CS + OA (0/4 h)	CS + ART (0/0 h)	CS + OA (0/0 h)	Match ID	Expression in A2780^cisR^/A2780	CS + ART (4/0 h)	CS + OA (0/4 h)	CS + ART (0/0 h)	CS + OA (0/0 h)
**1**	UR			PR	FUR	**48**	ND		OR		
**3**	DR	PR	PR	PR		**51**	DR	OR	PR	PR	PR
**4**	DR			PR	PR	**52**	ND	PR	PR	OR	
**5**	DR	PR	PR	PR		**54**	DR	OR	OR	OR	OR
**7**	DR			OR		**55**	DR			PR	PR
**8**	ND	PR	PR	OR	OR	**62**	DR				
**11**	ND	OR		PR		**63**	DR			PR	
**12**	ND	PR		PR		**66**	ND				
**13**	DR		PR	OR	OR	**68**	UR				
**14**	NC			OR	OR	**69**	DR			OR	OR
**15**	UR			OR	OR	**70**	ND			OR	
**16**	UR	PR		PR	PR	**74**	ND		OR	OR	
**17**	ND	OR	PR	OR	OR	**76**	UR				
**19**	DR	OR	PR	OR	OR	**78**	ND				
**20**	ND			OR	OR	**82**	ND				
**21**	UR				FUR	**85**	ND				
**25**	ND			OR	PR	**88**	ND				
**27**	ND	PR	OR	PR		**89**	ND			FR	OR
**29**	ND	OR				**94**	ND	OR	OR	OR	OR
**31**	ND		PR	OR		**95**	ND	OR			OR
**32**	ND		PR	OR	FR	**96**	ND			OR	OR
**33**	ND		PR	OR		**97**	ND	OR	OR		
**34**	ND		OR	OR		**98**	ND				
**35**	DR			PR	OR	**102**	ND				
**36**	UR		OR	OR	PR	**103**	DR		OR		
**39**	ND					**105**	ND				
**40**	DR					**106**	ND			OR	
**42**	ND		OR	OR		**108**	ND				
**43**	ND					**111**	ND		OR		
**44**	ND		OR			**116**	UR		PR		PR
**45**	DR	PR	FDR	FDR	FDR	**119**	ND			OR	
**46**	DR					**120**	ND		OR		
**47**	UR	OR	OR	OR	OR	**122**	ND				

FR: fully restored (the expression level of the protein after treatment was the same as its expression level in the untreated A2780 cell line). OR: over restored (means the value of protein expression after treatment was ± 0.05, as compared to its level in the untreated A2780 cell line); FDR: further down-regulated (the combined treatment had a negative effect on protein expression where the expression decreased at least 0.05, below the expression level compared to its level in the untreated A2780^cisR^ cell line); FUR: further up-regulated (the combined treatment has negatively affected the protein expression where the expression increased at least 0.05, above the expression level compared to its level in the untreated A2780^cisR^ cell line); PR: partially-restored; UP: up-regulated; DR: down-regulated; blank: spot was not detected after treatment with selected combination.

**Table 8 ijms-21-07500-t008:** List of protein expressions in, A2780^cisR^ compared to A2780 using A2780^cisR^ as reference.

Match ID	Expression in A2780^cisR^/A2780	CS + ART (4/0 h)	CS + OA (0/4h)	CS + ART (0/0 h)	CS + OA (0/0 h)
**8**	UR				
**9**	UR	OR	PR	PR	
**11**	DR	OR		OR	OR
**12**	UR		PR	OR	PR
**13**	UR	PR		OR	FR
**16**	ND				
**18**	ND	OR		PR	
**19**	DR		OR	OR	OR
**22**	UR	PR	PR	PR	FUR
**25**	UR			PR	
**26**	UR		FUR		FUR
**28**	DR	OR		OR	
**29**	ND			FUR	FUR
**30**	UR		OR	OR	PR
**32**	UR	FUR	PR	FUR	FUR
**41**	UR			PR	PR
**44**	DR			PR	OR
**46**	UR			PR	PR
**47**	UR			OR	
**50**	UR				
**56**	UR				
**57**	UR				
**58**	UR			PR	PR
**59**	DR				
**61**	UR		PR		
**62**	UR		FUR		
**63**	UR	PR	PR		
**69**	DR			OR	

FR: fully restored (the expression level of the protein after treatment was the same as its expression level in the untreated A2780 cell line); OR: over restored (means the value of protein’s expression after treatment is ± 0.05, as compared to its level in the untreated A2780 cell line); FDR: further down-regulated (the combined treatment had a negative effect on protein expression where the expression decreased at least 0.05, below the expression level compared to its level in the untreated A2780^cisR^ cell line); FUR: further up-regulated (the combined treatment has negatively affected the protein expression where the expression increased at least 0.05, above the expression level compared to its level in the untreated A2780^cisR^ cell line); blank: spot was not detected after treatment with selected combination; PR: partially-restored; UP: up-regulated; DR: down-regulated.

**Table 9 ijms-21-07500-t009:** Characteristics of proteins that have undergone differential expressions in A2780 and A2780^cisR^ cell lines following treatment with selected drug combinations using A2780^cisR^ cell line as reference.

Mach ID	Protein ID	Full Name	Mascot Search Results	Location	References
**8**	ATP5H O75947	ATP synthase subunit d, mitochondrial	Mass: 18480 Mascot score: 128 Coverage: 57% *pI*: 5.21 MS: 11 MSMS: 3	Mitochondrion	[15] http://www.matrixscience.com
**9**	RAN P62826	GTP-binding nuclear protein Ran	Mass: 24408 Mascot score: 132 Coverage: 39% *pI*: 7.01 MS: 11 MSMS:1	Nucleus	[15] http://www.matrixscience.com
**12**	PSA7 O14818	Proteasome subunit alpha type-7	Mass: 27870 Mascot score: 85 Coverage: 39% *pI*: 8.60 MS: 10 MSMS: 1	Cytoplasm	[15] http://www.matrixscience.com
**13**	TPI P60174	Triosephosphate isomerase	Mass: 30772 Mascot score: 340 Coverage: 49% *pI*: 5.65 MS: 20 MSMS:3		[15] http://www.matrixscience.com
**16**	VDAC1 P21796	Voltage-dependent anion-selective channel protein 1	Mass: 30754 Mascot score: 368 Coverage: 51% *pI*: 8.62 MS: 12 MSMS: 3	Mitochondrion outer membrane	[15] http://www.matrixscience.com
**18**	hnRNPA2/B1 P22626	Heterogeneous nuclear ribonucleoproteins A2/B1	Mass: 37407 Mascot score: 258 Coverage: 48% *pI*: 8.97 MS: 21 MSMS: 6	Nucleus, nucleoplasm	[15] http://www.matrixscience.com
**22**	PGK1 P00558	Phosphoglycerate kinase 1	Mass: 44586 Mascot score: 145 Coverage: 48% *pI*: 8.30 MS: 22 MSMS: 1	Cytoplasm	[15] http://www.matrixscience.com
**25**	GOT1 P17174	Glutamate oxaloacetate transaminase 1	Mass: 46219 Mascot score: 331 Coverage: 66% *pI*: 6.52 MS: 27 MSMS: 6	Cytoplasm	[15] http://www.matrixscience.com
**30**	EF1A1 P68104	Elongation factor 1-alpha 1	Mass: 50109 Mascot score: 135 Coverage: 25% *PI*: 9.10 MS: 13 MSMS: 2	Cytoplasm	[15] http://www.matrixscience.com
**32**	ACTG P63261	Actin, cytoplasmic 2	Mass: 41766 Mascot score: 598 Coverage: 55% *pI*: 5.31 MS:26 MSMS:5	Cytoplasm, cytoskeleton	[15] http://www.matrixscience.com
**41**	KPYM P14618	Pyruvate kinase PKM	Mass: 57900 Mascot score: 188 Coverage: 33% *pI*: 7.96 MS: 23 MSMS: 3	Cytoplasm.	[15] http://www.matrixscience.com
**44**	P4HB P07237	Prolyl 4-hydroxylase subunit beta	Mass: 57081 Mascot score: 433 Coverage: 50% *PI*: 4.76 MS: 29 MSMS: 6	Endoplasmic reticulum lumen	[15] http://www.matrixscience.com
**57**	APG2 P34932	Shock 70-related protein APG-2	Mass: 94271 Mascot score: 125 Coverage: 25% *pI*: 5.11 MS: 22 MSMS:3	Cytoplasm (Probable)	[15] http://www.matrixscience.com
**58**	VINC P18206	Vinculin	Mass: 123722 Mascot score: 226 Coverage: 34% *pI*: 5.50 MS: 40 MSMS:4	Cytoplasm, cytoskeleton	[15] http://www.matrixscience.com
**63**	EF2 P13639	Elongation factor 2	Mass: 95277 Mascot score: 425 Coverage: 27% *pI*: 6.41 MS: 39 MSMS:7	Cytoplasm	[15] http://www.matrixscience.com
**69**	CLIC1 O00299	Chloride intracellular channel protein 1	Mass: 26906 Mascot score: 158 Coverage: 47% *pI*: 5.09 MS: 11 MSMS: 3	Nucleus	[15] http://www.matrixscience.com

Accession number, protein ID, names and Mass, MSMS, coverage and protein score were obtained from APAF (http://www.proteome.org.au/). Theoretical isoelectric point (*pI*), subcellular location mass spectrum and matched peptides were obtained from Mascot database (http://www.matrixscience.com), Swiss-Prot database (http://www.uniprot.org/) [15]. Protein scores from Mascot database (http://www.matrixscience.com) through APAF (http://www.proteome.org.au/).

**Table 10 ijms-21-07500-t010:** Functional classification of the proteins that are differentially expressed in the A2780^cisR^ cell line compared to the A2780 cell line due to treatment using the parent cell line A2780 as reference.

Match ID	A2780^cisR^/A2780	Protein ID
Stress and Chaperones		
**3**	DR	CYPA
**32**	ND	TCPB
**35**	DR	ERp57
**39**	ND	TCPH
**45**	DR	HSP7C
**46**	DR	mortalin
**47**	UP	BIP
**51**	DR	HSP90B
**54**	DR	GRP94
**88**	ND	Hop
**105**	ND	TCPA
**Metabolism and Biosynthetic Processes**		
**11**	ND	PGAM1
**17/31**	ND	ENOA
**19**	DR	LDHB
**33**	ND	SERA
**55**	DR	IMMT
**63**	DR	NM23
**97**	ND	ATPA
**95**	ND	PSAT
**Cytoskeletal Proteins**		
**5**	DR	P18
**36/96**	UR	VIME
**94**	ND	GBLP
**108**	ND	CAH2
**Initiation and Elongation**		
**25**	ND	EFTU
**4**	DR	EIF5A1
**27**	ND	EF1G
**52**	ND	EF2
**mRNA Processing Proteins**		
**15**	UR	hnRNPA1
**16/76**	UR	hnRNP A2/B1
**Detoxification and Drug Resistance**		
**8**	ND	PRDX1
**66**	ND	PRDX6
**Cell Cycle Regulation and Cell Proliferation**		
**7**	DR	Op18
**89**	ND	MCM7
**Signal Transduction and Cell Cycle**		
**13**	DR	1433Z
**Protein Synthesis and Degradation**		
**69**	DR	PSA3

DR: Down-regulated; UR: Up-regulated; ND: Not-detected.

**Table 11 ijms-21-07500-t011:** Functional classification of the proteins that are differentially expressed in the A2780^cisR^ cell line compared to the A2780 cell line due to treatment using the A2780^cisR^ cell line as reference.

Match ID	A2780^cisR^/A2780	Protein ID
Stress and Chaperones		
**44**	DR	P4HB
**57**	UR	APG2
**Metabolism and Biosynthetic Processes**		
**13**	UR	TPI
**8**	UR	ATP5H
**25**	UR	GOT1
**16**	UR	VDAC1
**41**	UR	KPYM
**Cytoskeletal Proteins**		
**32**	UR	ACTG
**58**	UR	VINC
**Initiation and Elongation**		
**30**	UR	EF1A1
**63**	UR	EF2
**mRNA Processing Proteins**		
**18**	ND	hnRNP A2/B1
**Detoxification and Drug Resistance**		
**22**	UR	PGK1
**Cell Cycle Regulation and Cell Proliferation**		
**69**	DR	CLIC1
**Signal Transduction and Cell Cycle**		
**9**	UR	RAN
**Protein Synthesis and Degradation**		
**12**	UR	PSA7

DR: Down-regulated; UR: Up-regulated; ND: Not-detected.

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
