# Peer review of "Sequenced Combinations of Cisplatin and Selected Phytochemicals towards Overcoming Drug Resistance in Ovarian Tumour Models"

_ijms, 2020, doi:10.3390/ijms21207500_

Round 1
Reviewer 1 Report
The manuscript by Fazlul Huq and co-authors describe a study on the combination of several drugs in the treatment of ovarian cancer and extensive proteomic investigation of a shift in expression of proteins upon treatment. This results are very interesting and will be highly appreciated by the bioinorganic community. This manuscript can be accepted after minor revision. Authors should move most of the tables to the supplemental materials, especially table 9. Also, it is not necessary to describe a known data about each of the hit proteins just relevant reference will be sufficient.
Author Response
Please see in the attached

Reviewer 2 Report
The work of Alhuri et all investigates the combination of cisplatin and two phytochemicals that are oleanolic acid and artemisinin in two ovarian cell lines A2780 and the cisplatin resistant cell line A2780cis. This work is important from a pharmacological point of view and for the understanding of the resistance of A2780cis to cisplatin, effect that has been primary related to the efflux/influx cisplatin perturbation with respect to the A2780.
The discussion part is clear and focuses on the selected proteins which expression were initially found expressed differently in A2780 and A2780cis cell lines and then often modified when a synergistic effect has been observed between cisplatin and oleanolic or artemisinin.
However, the results part need however some improvements
Even if some details are in the experimental part, some of them should be included in the results to increase the comprehension of the results and allow believing consequently in the data.
General
- Line 65, introduce what are A2780cisR and A2780ZDD0473R cell lines and give some papers where they have been studies.
- Table I what is the treatment time?
Combination and Pt uptake
- Combination studies need to be more detailed in term of experimental procedures. We don’t clearly understand what is 0/0h…., the concentrations used for each combination, the number of combinations experiments allowing the CI values…, the time of treatment? One experimental graph should be shown to support the CI values given. Since the A2780cis is resistant is the dose used for combination is the same or higher than the one used for A2780?
- What are ED values? The ones of cisplatin, of the other drugs, of the combination assays? If the initial inhibition proliferation growth induced by one of the drug is already of 90%, how a combination effect could be evaluated?
- What are the Dm value, m values? They have been defined in tables 2, 3 and 4 but we do not understand their biological significance
- For clarity, figures 2 and 3 could be given together in one graph with histogram
- As well Figures 4 and 5 could be given in histogram. There is no connection between all the treatment, so the line representation is not the more appropriated.
- Concerning the Pt accumulation and amount of Pt bound to DNA quantification, what is the number of cells collected by pellets?
- Please, give the definition of AAS
- A lot of studies using ICP-mass for platinum uptake and quantification have been performed in both A2780 and A2780cis cell lines, could the authors cite some of these papers? line 275 p9.
- When comparing with the values of literature, it seems that the values of Pt for accumulation and bound to DNA found by the authors are 1000 times higher than the ones found in literature. What are the concentrations used? Could the authors check their values? At which treatment time combination treatment the quantification pf Pt has been performed?
Important point:
- The reviewer does not understand the reason why the proteomic studies have been performed following 24h treatments whereas the synergistic studies have been identified following 72h treatment
- Since both A2870 and A2780cis cells exist from long time ago, are the authors sure that the proteomic analysis of both cell lines has not been already performed? In literature it seems to be the case. If yes, what is different from this work?
- There are some “non English” words in the text and some mistakes where citing A2780 (A or one number missing)
Author Response
Please see in the attached
